# Self-assembled photonic cavities with atomic-scale confinement

Ali Nawaz Babar[1,2 ✉], Thor August Schimmell Weis[1], Konstantinos Tsoukalas[1], Shima Kadkhodazadeh[2,3], Guillermo Arregui[1], Babak Vosoughi Lahijani[1,2] & Søren Stobbe[1,2 ✉]

Despite tremendous progress in research on self-assembled nanotechnological building blocks, such as macromolecules[1], nanowires[2] and two-dimensional materials[3], synthetic self-assembly methods that bridge the nanoscopic to macroscopic dimensions remain unscalable and inferior to biological self-assembly. By contrast, planar semiconductor technology has had an immense technological impact, owing to its inherent scalability, yet it seems unable to reach the atomic dimensions enabled by self-assembly. Here, we use surface forces, including Casimir–van der Waals interactions[4], to deterministically self-assemble and self-align suspended silicon nanostructures with void features well below the length scales possible with conventional lithography and etching[5], despite using only conventional lithography and etching. The method is remarkably robust and the threshold for self-assembly depends monotonically on all the governing parameters across thousands of measured devices. We illustrate the potential of these concepts by fabricating nanostructures that are impossible to make with any other known method: waveguide-coupled high-$Q$ silicon photonic cavities[6,7] that confine telecom photons to 2 nm air gaps with an aspect ratio of 100, corresponding to mode volumes more than 100 times below the diffraction limit. Scanning transmission electron microscopy measurements confirm the ability to build devices with sub-nanometre dimensions. Our work constitutes the first steps towards a new generation of fabrication technology that combines the atomic dimensions enabled by self-assembly with the scalability of planar semiconductors.

The fabrication of functional materials and devices at the micro- and nanoscale typically follows either a top-down approach based on planar technology or a bottom-up approach, where structures are self-assembled using various effects, such as van der Waals, electrostatic, capillary or hydrogen-bonding forces[8–12]. While top-down nanofabrication underpins the scalability of semiconductor technology, the bottom-up approach has enabled a wide range of research on devices with near-atomic dimensions. Such miniaturization is crucial for a wealth of research and technology that rely on an increased surface-to-volume ratio, strong field gradients or quantum effects. However, the miniaturization of semiconductor technology has slowed, and the current industry roadmap forecasts no lateral lithography features (minimum half-pitch or physical gate length) below 8 nm until 2037. At the same time, while bottom-up approaches can achieve feature sizes down to atomic scales, synthetic self-assembly remains far from capable of replicating the hierarchical and scalable self-assembly in biological systems[13]. A practical consequence is that a wealth of research on bottom-up nanotechnology for information technology always had to rely on top-down technology for the interconnect architecture. For example, lithographically defined wires or waveguides are needed to contact single-molecule

devices[14] or single-quantum-dot devices[7]. Combining the scalability of top-down planar technology with the resolution of bottom-up approaches would open vast perspectives[15], but they are commonly considered disjoint. Self-assembly has been explored for fabricating origami-like structures in microelectromechanical systems[16], which enable unique geometries but not nanoscale dimensions. Strategies for combining nanoscale self-assembly with planar technology are therefore scarce[17,18], and a pathway for their direct integration was so far missing.

Recent developments have brought miniaturization to the centre stage also in photonics because it is required for realizing dielectric bowtie cavities[5] with ultrasmall mode volumes—a regime previously believed to be accessible only in plasmonics[19,20]. The existence of dielectric cavities with mode volumes below the diffraction limit was predicted[21] in 2005 but demonstrated only very recently[5], in part because realistic designs were missing and in part because experimental progress was impeded by the extreme requirements on the nanofabrication. Dielectric bowtie cavities harness the field discontinuities at material boundaries to strongly confine light inside dielectrics[22,23], and hold the promise of unprecedented light–matter interaction strengths, fostering new developments in nanolasers and optical interconnects[24],

[1]DTU Electro, Department of Electrical and Photonics Engineering, Technical University of Denmark, Kongens Lyngby, Denmark. [2]NanoPhoton - Center for Nanophotonics, Technical University of Denmark, Kongens Lyngby, Denmark. [3]DTU Nanolab, Technical University of Denmark, Kongens Lyngby, Denmark. ✉e-mail: anaba@dtu.dk; ssto@dtu.dk

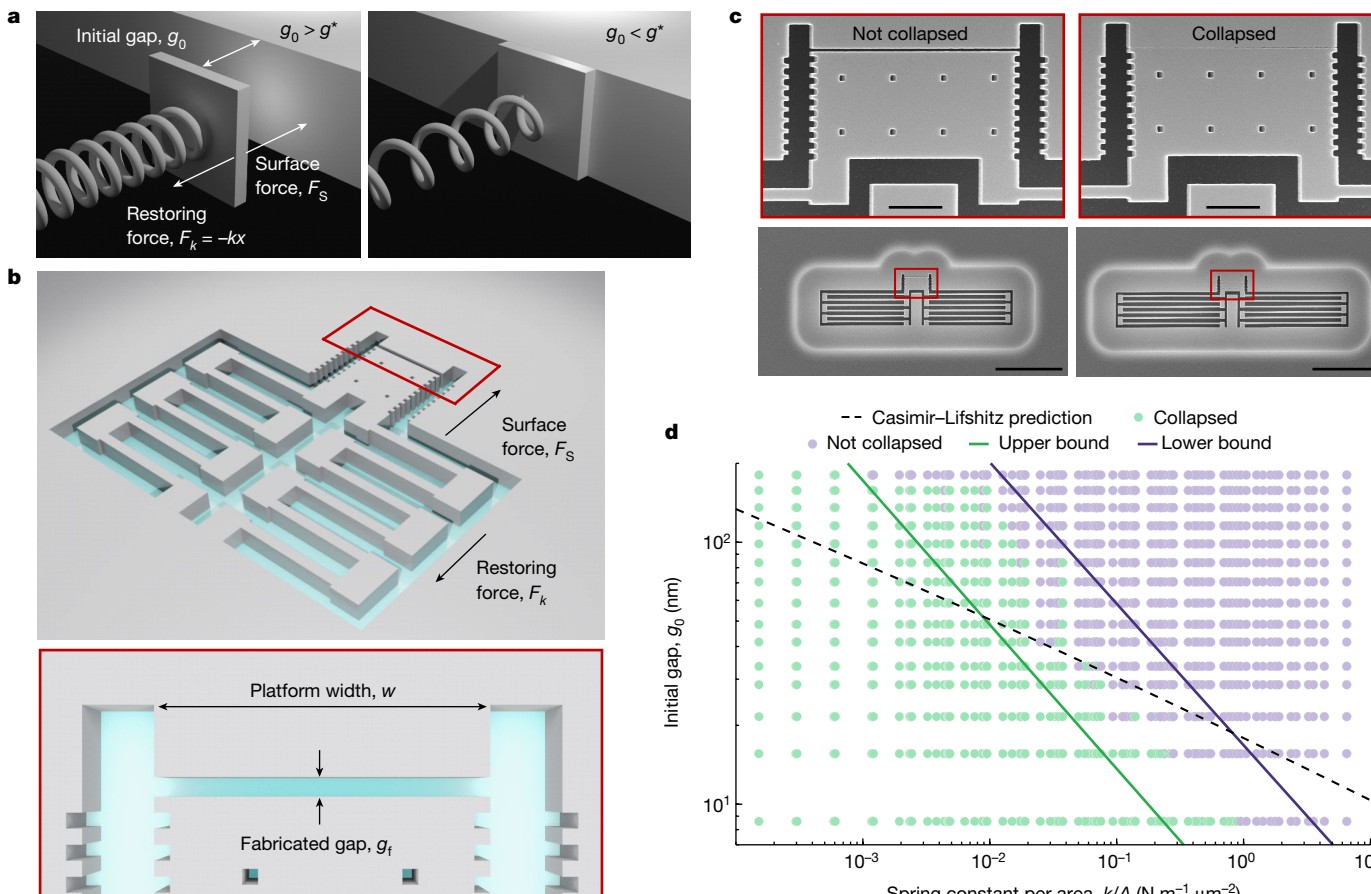

**Fig. 1 | Deterministic in-plane self-assembly of suspended silicon platforms by surface forces. a**, Experimental concept for mapping the design space for self-assembly exploiting the pull-in instabilities associated with on-chip surface forces. The balance between the nonlinear surface forces and the linear spring forces means that devices with initial gaps, $g_0$, that are larger (smaller) than the critical gap, $g^*$, do not collapse (collapse deterministically) as shown on the left (right). **b**, Illustration of the realization of the experiment in a silicon-on-insulator platform before release by underetching the silicon device layer. The force balance depends on the platform width, $w$, the initial gap, $g_0$, and the spring constant, $k$. The small difference between the fabricated and initial gaps stems from stress release after underetching, which is taken into account in the experiments. **c**, Tilted-view (20°) SEM images of two released devices with $w = 4$ μm, $g_0 = 41$ nm and either $k = 0.038$ N m$^{-1}$ (left, not collapsed) or $k = 0.019$ N m$^{-1}$ (right, collapsed). **d**, Measured map of the design space for self-assembly with compliant silicon structures, obtained by characterizing 1,536 platforms by SEM. The green circles represent the collapsed platforms, and the purple represent the non-collapsed platforms. All the devices below the upper bound collapse, while those above the lower bound do not. Scale bars, 1 μm (**c**, top); 10 μm (**c**, bottom).

nonlinear photonics[25], all-optical switching[26], cavity quantum electrodynamics[7] and cavity optomechanics[27]. The width of the bowtie is the most crucial parameter because it determines the field enhancement[22]. The first experiment[5] demonstrating confinement of light below the diffraction limit in dielectric bowtie cavities employed 8 nm wide silicon bridges with an aspect ratio of 30, and although minor improvements along this route may be possible, it appears futile to try to scale conventional lithography and etching to atomic dimensions with aspect ratios exceeding 100. Void or low-refractive-index features with extreme aspect ratios are especially challenging to fabricate, but they are required for some of the most radical applications of nanocavities, such as bulk non-linearities operating at the single-photon level[25] and single-photon emitters for quantum photonic integrated circuits[28]. While top-down nanofabrication in modern foundry production allows defining the position of edges with both precision and accuracy (disorder) at the level of a single silicon atom[29], the resolution limit (the smallest possible linewidth) of approximately 50 atoms has so far hindered such developments, see Supplementary Information Section 1.

Here we propose and demonstrate a new approach to the manufacturing of semiconductor devices. It allows building devices with unprecedented dimensions by using the ubiquitous surface forces that act on objects separated by a few tens of nanometres, such as the Casimir-van der Waals force[4,30]. While conventionally considered nuisances that cause the failure of micro- and nanomechanical devices[31], our experiments harness these forces to enable controlled, deterministic and directional collapses to fabricate nanostructures with few- or even sub-nanometre dimensions.

## Deterministic self-assembly

Most approaches to self-assembly rely on liquid suspensions[9,12], but our method exploits the intrinsic pull-in instabilities occurring between two nearby objects when a surface force, $F_s$, is countered by a linear restoring spring force, $F_k$. This is illustrated in Fig. 1a: when the gap equals a critical gap, $g^*$, the pull-in instability triggers a collapse, and the objects subsequently adhere to each other by van der Waals forces, resulting in a structurally stable self-assembled device. Although the surface forces are well understood theoretically[4,32], their exact numerical values are difficult to determine because they depend strongly on parameters such as surface treatment, doping level and fabrication imperfections[33]. Therefore, the starting point of our investigation is to

map the surface-force instability as a function of geometry, thus providing design rules for self-assembly by directed collapses. This experiment is implemented in a silicon-on-insulator platform, as illustrated in Fig. 1b, using 220-nm-thick suspended silicon platforms close to a rigid and anchored silicon structure. The silicon platforms are attached to the frame by two symmetric folded cantilever springs of spring constant $k$ and separated from the anchored part by a gap, $g_f$. Our devices are defined using electron-beam lithography and reactive-ion etching. Subsequently, the platforms are released from the substrate by selective etching of the oxide layer using anhydrous vapour-phase hydrofluoric acid. The released platforms collapse in-plane onto the anchored structure if the initial gap, $g_0$, is smaller than the critical gap, $g^*$, which in turn depends on $k$ and the platform width, $w$. This is illustrated in Fig. 1c, which shows representative scanning electron microscope (SEM) images of two devices with the same $w$ and $g_0$, but different $k$ (see Supplementary Information Sections 2.1 and 2.2 for details on the extraction of $k$ and $w$ as well as the relation between $g_0$ and $g_f$). The stiffer spring provides enough restoring force for the platform to reach a stable equilibrium at a small displacement without collapsing, while that with a smaller spring constant does not, leading to a deterministic and directed collapse.

To map out the geometry dependence and reproducibility of the pull-in instability threshold, we fabricate 2,688 devices with varying values of $w$, $k$ and $g_0$ distributed across two samples, A and B. We perform systematic SEM characterization of all devices after under-etching and record which structures collapse and which do not. The resulting data for sample A is shown in Fig. 1d. We identify two gaps for fixed values of $w$ and $k$: the largest value of $g_0$ for which the collapse occurs and the smallest value of $g_0$ for which the collapse does not occur. We find that all platforms for which $g_0 < 3.8 \times (k/A)^{-0.55}$ collapse and all platforms for which $g_0 > 16.8 \times (k/A)^{-0.54}$ do not collapse, where $A = wh$ and $h = 220$ nm is the device-layer thickness. We note that the largest initial gap leading to collapse and the smallest gap not leading to collapse are adjacent data points across our entire data set of 2,688 devices, and only 11 devices failed due to out-of-plane collapse and/or lithographic errors. The collapse threshold provides the essential design rule for realizing suspended silicon devices with high-aspect-ratio gaps that avoid unintended pull-in instabilities, such as nano-opto-electromechanical systems[34], or, in the opposite limit, the criterion for deterministic self-assembly. Even if our static collapse experiment does not aim to replicate the abundance of accurate dynamical measurements of the Casimir–van der Waals force available in the literature[4,31,32,35], we include in Fig. 1d the critical gap calculated with the Lifshitz theory of the Casimir–van der Waals force in the proximity force approximation (PFA) for two silicon slabs. We observe good agreement with the measured collapse threshold in the range where the PFA is expected to be valid, that is, for gaps in the range of 20 to 50 nm. The platforms are found to be more prone to collapse than predicted by the model for larger initial gaps and smaller spring constants, which indicates additional attractive contributions to the net surface force, such as electrostatic surface effects[36] and effects beyond the PFA[37]. We refer to Supplementary Information Sections 2.3 and 2.4 for raw data, details on the theoretical model and further discussion of its validity.

## Self-assembly of atomic-scale cavities

To illustrate the application of our method, we now turn to the realization of photonic nanocavities that confine light in air gaps in silicon membranes with aspect ratios exceeding 100. Figure 2a shows the geometry of a nanobeam photonic-crystal cavity featuring a unit cell that includes a 2 nm air bowtie. The normalized electric field of the fundamental optical mode is plotted respectively in Fig. 2b and, zooming-in on the central bowtie, in Fig. 2c. The fundamental cavity mode features a resonance wavelength of $\lambda = 1,524$ nm, a quality factor

of $Q = 5 \times 10^4$ and a mode volume of $V = 3.36 \times 10^{-4} \times \lambda^3$, calculated at the centre of the central bowtie[22]. The cavity design deliberately violates the constraints of our nanofabrication process[5] by including 2 nm air voids at the bowtie centres, and it is therefore realized following the design rules provided by Fig. 1d (see Supplementary Information Section 3 for details on the cavity design).

The cavity is fabricated as two spring-suspended nanobeams separated by a gap. A characteristic lithographic mask for the nanobeam geometry is shown in Fig. 2d, where we introduce an offset, $\delta$, between the gap that promotes the collapse, $g_P$, and the tip-to-tip gap, $g_T$. Importantly, the resolution of the fabrication process puts a lower bound to the value of the fabricated gap, $g_f$, and to the radius of curvature of sharp features, $R_S$, when the pattern is transferred onto the silicon device (Fig. 2e). However, the relative difference between the tip-to-tip distance before self-assembly, $g_b$, and $g_f$, can be made arbitrarily small by adjusting the electron dose or etching parameters. In our case the bowtie width, $g$, is controlled by $\delta$, as evidenced by a systematic SEM study of devices fabricated with varying $\delta$ (Supplementary Information Sections 4.2 and 4.3. This enables the realization of single-digit-nanometre air bowties, for example, approximately 2 nm for offset $\delta = 10$ nm (Fig. 2f). Figure 2g shows a characteristic device, which includes the two pairs of folded cantilever springs. However, few- or sub-nanometre gaps cannot be reliably measured with SEM. Therefore, we turn to characterization using scanning transmission electron microscopy (STEM), details of which are found in the Methods and Supplementary Information Section 4.4. Figure 2h,i show top-view annular dark-field STEM images of the central bowtie unit cell for self-assembled nanobeam cavities fabricated using $\delta = 10$ nm and $\delta = 11$ nm. We tilt the sample to align the electron beam to the [100] zone axis of the silicon membranes, revealing the (022) planes of silicon. We find the (022) crystal planes in the top and bottom parts of the bowties to be misaligned by 1–2°, for example, 1.6° and 1.8° respectively in Fig. 2h,i. We attribute the observed angles to minor deviations from the perfect sidewall verticality (Supplementary Information Section 4.1), which only has a minor effect on the resulting devices because the compliant halves simply bend by the same angle out-of-plane while keeping the interfaces parallel. We therefore acquire the STEM images by keeping the bottom half of the bowtie normal to the incident electron beam, as shown in Fig. 2h,i. In Fig. 2h, the silicon (022) crystal plane of the upper and lower parts of the self-assembled bowtie are separated by a distance of 9.2 nm. The air bowtie is bounded on both sides by amorphous silicon oxide between the two crystalline regions, as confirmed by atomic composition analysis using electron energy-loss spectroscopy. From the STEM data, we estimate the thickness of the amorphous layer, $d$, to be in the range of 2 to 2.5 nm, which confirms that it is the native silicon oxide. Due to the small tilt angle of the top part of the nanobeam cavity, the STEM shows a smooth transition from the background through the oxide to silicon, while the lower part shows a sharper transition. The high-resolution STEM image of the central part of the bowtie for $\delta = 11$ nm, shown in Fig. 2i, indicates that the two bowties are most likely touching at the native-oxide interface. This demonstrates the ability of our method to build atomic-scale semiconductor devices in which the smallest critical dimension is limited by structural disorder rather than resolution.

## Integration with photonic circuits

We characterize the resonant modes of the self-assembled nanobeam cavities by cross-polarized far-field resonant scattering, which results in Fano lineshapes due to interference with a vertical mode[5] of the structure (Fig. 3a). We extract the resonant wavelengths and quality factors of the fundamental cavity mode in a series of cavities fabricated with $\delta = 8$, 9 and 10 nm. We estimate the bowtie widths (average and standard deviation) to be $g = 5.0 \pm 1.8$, $3.0 \pm 1.8$ and $1.0 \pm 1.8$ nm, respectively, from interpolation using an SEM-extracted offset-to-width correspondence

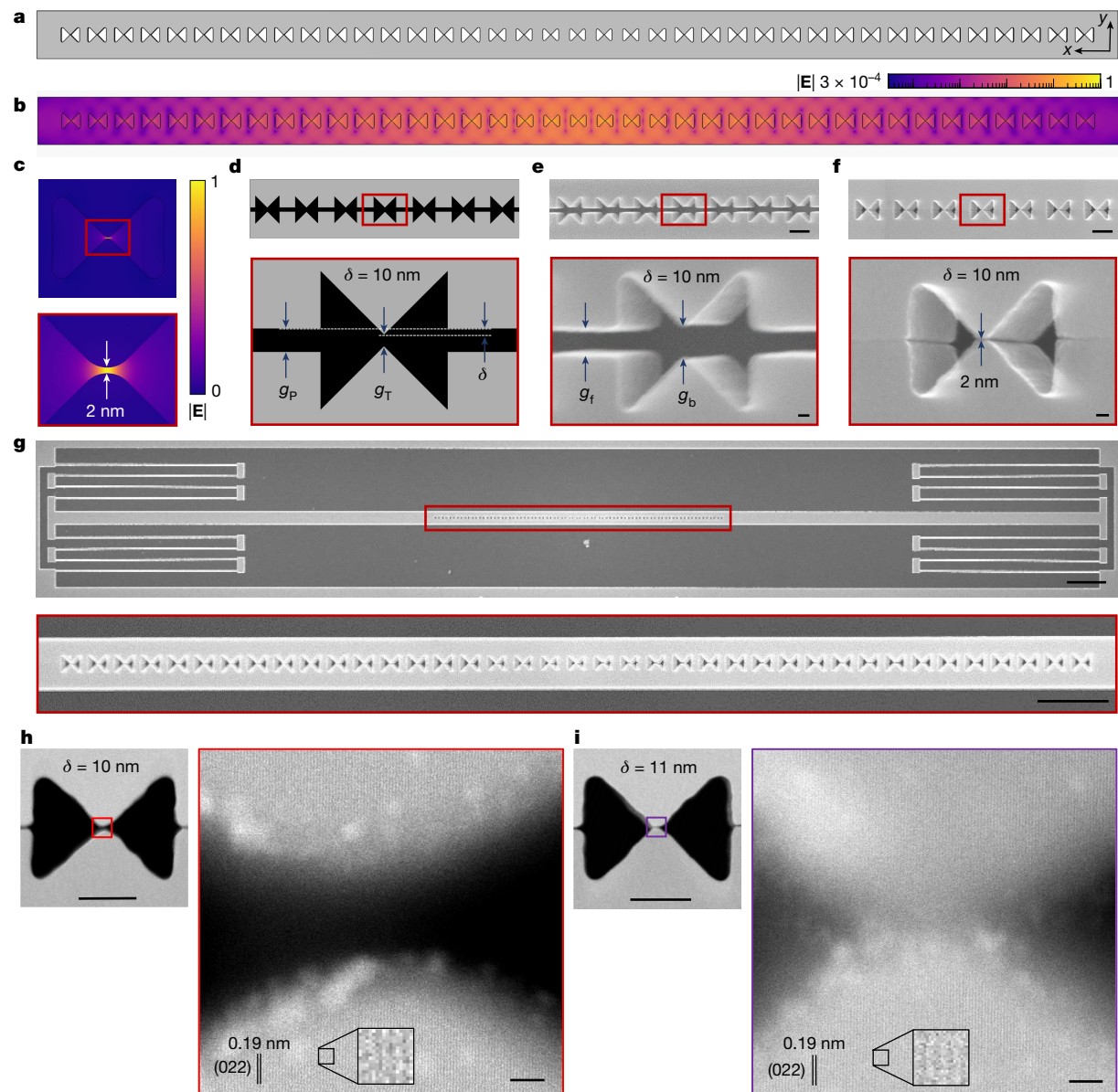

**Fig. 2 | Design and fabrication of self-assembled silicon nanobeam bowtie cavities. a**, Geometry of the nanobeam bowtie cavity. **b**, Normalized electric field of the cavity mode, |**E**|, with a logarithmic colour map. **c**, Normalized electric field, |**E**|, of the central bowtie unit cell of the nanobeam cavity, with a linear colour map. **d**, Electron-beam-lithography mask schematic of the central region of the nanobeam cavity, with black (grey) illustrating exposed (unexposed) areas. The offset, $\delta$, is defined as $\delta = (g_P - g_T)/2$. **e**, Tilted (20°) SEM image of the central part of the cavity before the release etch triggers the self-assembly of the two parts initially separated by $g_f = 50$ nm, except at the

bowtie where the distance is $g_b = 52$ nm. **f**, Tilted (20°) SEM of the central part of a nanobeam cavity after self-assembly, with the approximately 2 nm gap indicated in the zoom-in. **g**, Top-view SEM image of the full device, including the spring suspension. **h,i**, Top-view high-resolution STEM images of the central unit cell and their bowtie tips for cavities fabricated using $\delta = 10$ nm (**h**) and $\delta = 11$ nm (**i**). The scale bars are calibrated using the 0.19 nm interplanar distance of the visible (022) crystalline silicon planes. Scale bars, 200 nm (**e,f**, top); 20 nm (**e,f**, bottom); 2 µm (**g**, top); 1 µm (**g**, bottom); 100 nm (**h,i**, left); 2 nm (**h,i**, right).

(Supplementary Information Section 4.3). In addition, we simulate the fabricated geometry, which includes a 2 nm native-oxide layer, for varying $g$ and compare the simulated and measured characteristics. The theory curve in Fig. 3b evidences a pronounced red-shift with decreasing bowtie width, which is confirmed by the experimental data set despite the large wavelength variance at fixed width. The measured $Q$-factors, which are found to be between $7.54 \times 10^3$ and $4.21 \times 10^4$, grow with bowtie width both in the experiment and simulation, as shown in Fig. 3c. We attribute the approximately 4-fold reduction in $Q$ to scattering losses due to structural disorder. Still, we consistently observe $Q$-factors exceeding previous experimental results on subdiffraction confinement by more than an order of magnitude across multiple

devices, even for few-nanometre cavities that exhibit much smaller mode volumes than any previous experiments on dielectric cavities[5,22] (Fig. 3b). We refer to Supplementary Information Sections 3.3, 3.4, 3.5 and 5.1 for details on the simulations, the full experimental data set and their comparison.

Finally, we turn to the quest of interfacing and self-aligning self-assembled devices with complex circuitry and, more generally, to explore the scalability of our method for interfacing the bottom-up self-assembled devices with top-down planar technology. To this end, we use a recently invented topology-optimized photonic coupler that enables a broadband waveguide-to-waveguide transmission window across a 100 nm air trench, providing electrical and

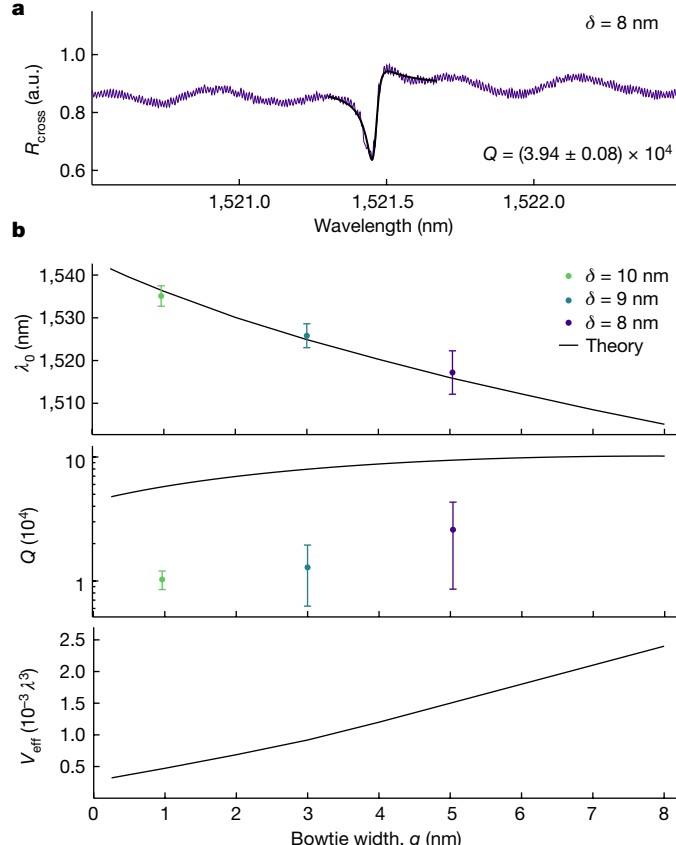

**Fig. 3 | Resonant scattering for self-assembled nanobeam cavities.**
**a**, Representative cross-polarized scattering spectrum, $R_{cross}$ (purple line), of a cavity fabricated with $\delta = 8$ nm. The fitted Fano lineshape over the fitting range is included (black line) and the extracted quality factor, $Q$, is given. **b**, Simulated and measured resonant wavelengths, $\lambda_0$, quality factors, $Q$, and effective mode volumes, $V_{eff}$, as a function of the bowtie width, $g$. The simulations use all measured dimensions, including a 2 nm native-oxide layer. The theory curve for $\lambda_0$ is red-shifted by 13 nm due to the variations in the device-layer thickness and the SEM-extracted dimensions. The experimental bowtie widths are estimated following Supplementary Information Section 4.3, and the vertical error corresponds to the standard deviation obtained from measurements on a set of nominally identical cavities.

mechanical isolation[38]. A self-assembled nanobeam cavity, including efficient interfaces to external waveguides via such couplers and low-loss anchor points for the springs on tapered waveguide regions, is shown in Fig. 4a. Notably, the trench in the coupler enables the use of the self-assembly method by fabricating the coupler in two halves (Fig. 4b), which self-assembles at the same time as the nanobeam cavity (Fig. 4c). Still, some out-of-plane bowing is observed, which could readily be avoided by adding more springs or other means of stress-release management. As in the structures in Fig. 2, two sets of springs are used, but they are attached to the tapered waveguide section as shown in Fig. 4d. Compared to the cavity shown in Fig. 2, the cavity for on-chip transmission experiments, which is shown in Fig. 4e, has a longer defect region to reduce out-of-plane radiation losses and a smaller number of mirror unit cells to facilitate efficient transmission through the cavity. We measure the wavelength-dependent transmitted power through the self-assembled circuits and normalize the acquired spectra to that measured in a self-assembled suspended waveguide of equivalent length. Figure 4f shows the transmittance, $T$, for two different self-assembled devices: first, a 2 nm air-bowtie nanobeam mirror with 25 identical unit cells, and second, a 2 nm air-bowtie nanobeam cavity with 8 mirror unit cells, both corresponding to

structures with $\delta = 12$ nm. For the employed sample, $\delta = 12$ nm corresponds to a 2 nm bowtie width instead of the value of $\delta = 10$ nm as demonstrated for the cavity in Fig. 2 (see Supplementary Information Section 4.2 for details on the variation). The transmittance of the mirror is negligible between 1,425 nm and 1,540 nm, which is consistent with the simulated photonic band gap. The spectrum of the cavity device exhibits three distinct Lorentzian resonances, which agree quantitatively with our numerical cavity model using the fabricated dimensions. We observe a 41% cavity transmittance, which is smaller than the simulated value of 96.4% due to structural disorder. The transmittance across the fundamental cavity mode is shown in Fig. 4g, and exhibits an irregular lineshape due to interference from reflection at the grating couplers. By fitting to a Lorentzian lineshape, we obtain a $Q$ of $(1.44 \pm 0.02) \times 10^4$, which is comparable to the values obtained in Fig. 3 but with the notable difference that this is a loaded $Q$-factor and the cavity is efficiently coupled to a waveguide architecture. We also observe some fluctuations of $Q$ and $T$ in nominally identical self-assembled waveguide-coupled cavities (Supplementary Information Section 5.2).

## Conclusion

Our mapping of the phase space governing the collapse of suspended platforms provides a clear design rule both for new research aiming to exploit the deterministic self-assembly and for conventional micro- and nanoelectromechanical systems, where collapses are generally undesirable. The ubiquitous nature of surface forces implies that our concepts can be applied to any material, which may even be coated with ultra-thin functional materials, for example, using atomic-layer deposition, which would then be embedded in ultra-compact devices. Our demonstration of optical cavities with few- to sub-nanometre features therefore takes the first steps towards a new generation of integrated nanophotonic devices exhibiting extreme field intensities that may approach both the limits to light–matter interaction strength[39] and regimes where the continuum model of electromagnetism breaks down[40]. Such levels of confinement may impact a wide span of potential applications such as surface-enhanced Raman spectroscopy[17], nonlinear optics[25], biosensing[41] and quantum technologies[7,42]. For example, our self-assembled waveguide-coupled cavity features an unprecedented set of parameters: with a mode volume of $8.8 \times 10^{-4}$ cubic wavelengths and a loaded $Q$-factor of $1.4 \times 10^4$, the light–matter interaction is enhanced by a Purcell factor of $1.3 \times 10^6$ with a wide bandwidth and a high on-resonance transmission. By incorporating embedded emitters such as erbium-doped alumina[43], highly efficient single-photon sources at telecom wavelengths may be envisioned, possibly even with a high degree of quantum coherence due to the extreme Purcell enhancements. Such cavities may also enhance the bulk nonlinearity of embedded materials to a level where they could operate using single photons[25] and provide record single-photon optomechanical readout rates for gigahertz mechanical modes even in the absence of embedded materials[27]. While our work showcases the self-assembly of photonic cavities with few- to sub-nanometre confinement, our method may be applied in a much broader field of research and technology, for example, solid-state nanopore sequencing[44], nanogap quantum tunnelling electrodes[41] or ultra-high-quality shadow masks for superconducting quantum electronic devices[45]. More generally, our work opens perspectives for exploring new regimes of photonics, electronics and mechanics at atomic scales while at the same time enabling scalable and self-aligned integration with large-scale chip architectures.

## Online content

Any methods, additional references, Nature Portfolio reporting summaries, source data, extended data, supplementary information,

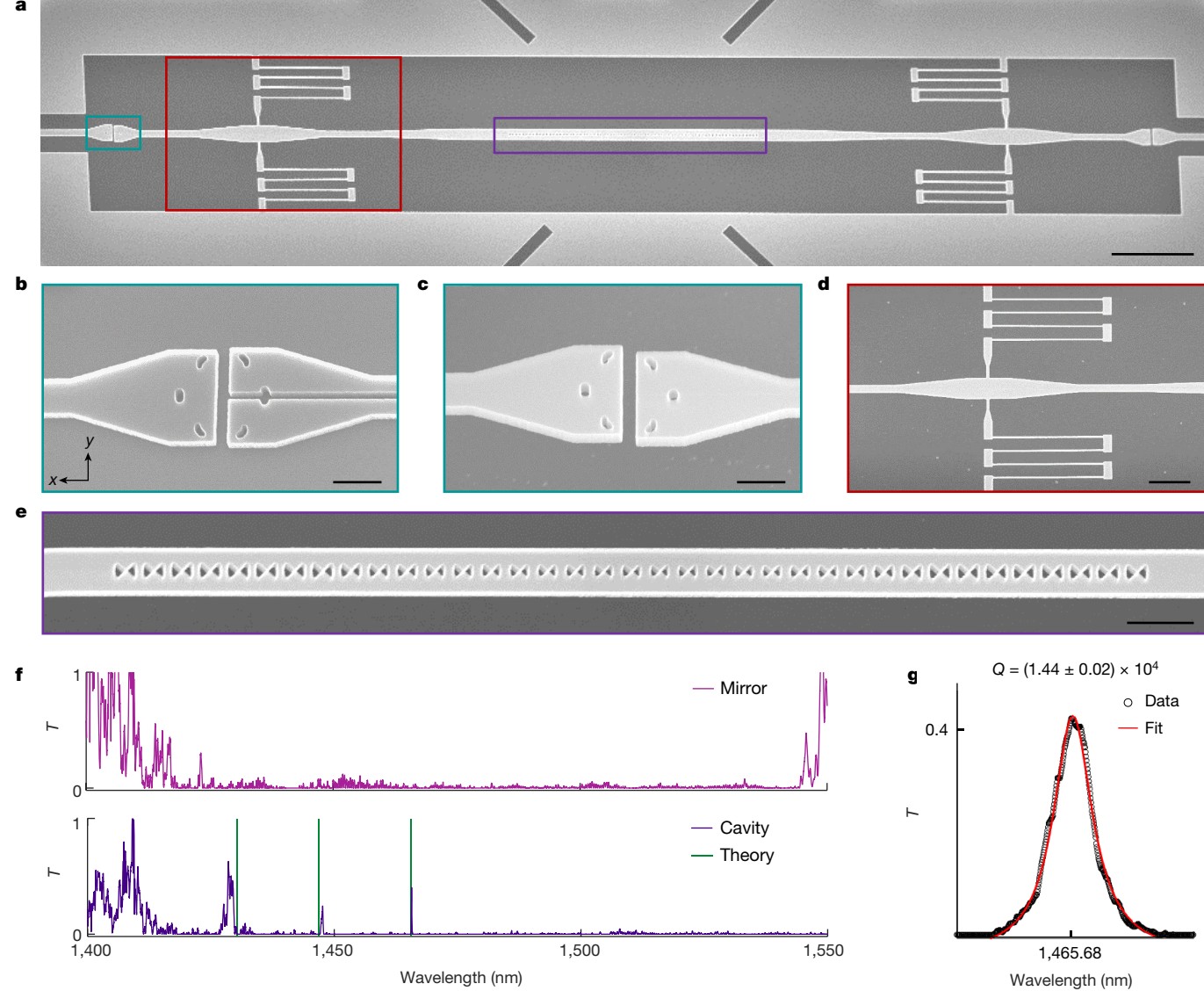

**Fig. 4 | Integration of self-assembled nanobeam cavities with photonic circuits. a**, Tilted-view (20°) SEM image of a self-assembled nanobeam cavity terminated with air-trenched waveguide-to-waveguide couplers and including tapered waveguide sections and suspension springs. **b,c**, Tilted-view (15°) SEM images of a trenched coupler before (**b**) and after (**c**) self-assembly. **d**, Top-view SEM image of the spring suspension attached to a tapered waveguide section. **e**, Tilted-view (20°) SEM image of a self-assembled nanobeam cavity with a 2 nm bowtie width. **f**, Transmission spectrum of a self-assembled nanobeam bowtie mirror (top) and cavity (bottom) normalized to that of a reference structure based on a straight self-assembled waveguide. The resonant wavelengths of the simulated modes are represented with vertical green lines for reference. The simulated modes are red-shifted by 11.6 nm and well within the differences between experiments and theory commonly observed in nanophotonic devices fabricated with top-down nanopatterning. **g**, Lorentzian fit to the fundamental cavity resonance. Scale bars, 5 μm (**a**); 500 nm (**b,c**); 2 μm (**d**); 1 μm (**e**).

acknowledgements, peer review information; details of author contributions and competing interests; and statements of data and code availability are available at https://doi.org/10.1038/s41586-023-06736-8.

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

## Methods

### Fabrication process

The devices are fabricated on a commercial silicon-on-insulator substrate (Soitec) with a 220 nm silicon device layer and a 2 μm buried oxide layer. A two-layer hard mask is deposited on the silicon device layer, consisting of 30 nm poly-crystalline chromium and 12 nm poly-crystalline silicon layers, followed by a 50 nm layer of chemically semi-amplified resist (CSAR) applied by spin-coating. The patterns are exposed in the resist with a 100 keV 100 MHz JEOL9500FSZ electron-beam writer and transferred into the silicon device layer by a low-power switched reactive-ion etch. The buried oxide layer is selectively etched to suspend the devices with an anhydrous hydrofluoric-acid (99.995%) vapour phase etcher (SPTS Primaxx uEtch), using ethanol as a catalyst. A process pressure of 131 Torr, and a slow etching recipe (etch rate of approximately 14 nm min$^{-1}$) are used for selective oxide etching. The fabrication process flow is detailed in ref. 5, and the hard-mask etching process is detailed in refs. 46,47.

### Surface-force characterization

The measurements in Fig. 1 are performed on sample A (1,536 devices) with platforms of widths $w$ = [2, 3, 4, 5, 8, 16, 32, 64] μm with logarithmic variations of both $g_f$ and $k$ from 30 to 200 nm and from 0.0097 to 13 N m$^{-1}$, respectively. The silicon-on-insulator stack sets the thickness of the platform and height above the substrate to $h$ = 220 nm and $H$ = 2 μm, respectively. An array of squares ($w \times 2$) with a sidelength of 200 nm and a pitch of 1 μm is etched in the silicon platform to facilitate the underetching. The devices also have trenches on the top-right and top-left of the platform to reduce potential fringing-field contributions to the surface forces[48]. Scales are integrated on the right and left side of the platforms to measure displacements due to the built-in stress release, which imposes a baseline correction to the initial gap $g_0$ (Supplementary Information Section 2.1). All these additional features have minimal effect on whether the devices collapse or do not collapse.

### Scanning transmission electron microscopy

Annular dark-field STEM imaging is performed using an FEI Titan 80-300 kV transmission electron microscope (TEM) operated at 300 kV to extract high-resolution images of the cavity bowtie region. The transmission electron microscope is fitted with a field-emission gun and an aberration-correction unit on the probe-forming lenses, giving it a spatial resolution better than 0.1 nm. A focused ion beam is used inside a FIB-SEM system (Helios Nanolab 600) to prepare the cavity structures for high-resolution imaging. A micromanipulator needle is welded to the cavity structure by induced deposition of Pt from a precursor source to transport the cavities from the sample to the TEM equipment. This is followed by cutting the tethers around the cavities using a Ga+ ion beam of 30 keV and 40 pA current, lifting the released cavities from the sample, relocating and welding the cavities to a TEM-compatible Cu grid, and finally detaching the micromanipulator needle from the cavities using the ion beam.

### Optical measurements

The optical spectrum of each nanocavity is measured using free-space confocal microscopy. Measurements are performed either by direct resonant scattering on isolated nanocavities (Fig. 3) or via transmission by coupling light in and out of photonic circuits with embedded nanocavities (Fig. 4). Two fibre-coupled tuneable diode lasers (Santec TSL-710, $\lambda_1$ = 1,355–1,480 nm and $\lambda_2$ = 1,480–1,640 nm) are combined into a 4 × 1 optical switch (Santec OSU-110) for excitation of the nanocavities. Light is focused onto and collected from the sample using a microscope objective (Mitutoyo Plan Apo NIR 50X, numerical aperture = 0.42). For resonant scattering measurements, the excitation and collection spots overlap, and their polarizations are set at 45° relative to the leading polarization of the cavity mode and orthogonal to each other (see Supplementary Information Section 5.1 for further details). For measurements of nanocavities embedded in photonic circuits, the excitation and collection are cross-polarized and spatially offset by employing two free-space grating couplers oriented orthogonal to each other. The grating couplers are kept 30 μm apart in both the vertical and the horizontal direction. Spectra are acquired by sequentially sweeping the two tuneable lasers (if needed) and detecting with a synchronous calibrated power-metre (Santec MPM-210). The spectra are then normalized to the laser spectrum as measured with a direct patch fibre for resonant scattering measurements and to the spectrum of a suspended silicon waveguide of equivalent length for the photonic circuits.

## Data availability

The data used in this paper are available at https://doi.org/10.5281/zenodo.8301463 (ref. 49).

## Code availability

The code used in this paper is available at https://doi.org/10.5281/zenodo.8301463 (ref. 49).

46. Arregui, G. et al. Cavity optomechanics with Anderson-localized optical modes. *Phys. Rev. Lett.* **130**, 043802 (2023).
47. Rosiek, C. A. et al. Observation of strong backscattering in valley-Hall photonic topological interface modes. *Nat. Photon.* **17**, 386–392 (2023).
48. Tsoukalas, K., Vosoughi Lahijani, B. & Stobbe, S. Impact of transduction scaling laws on nanoelectromechanical systems. *Phys. Rev. Lett.* **124**, 223902 (2020).
49. Babar, A. N. et al. Self-assembled photonic cavities with atomic-scale confinement. *Zenodo* https://doi.org/10.5281/zenodo.8301463 (2023).

**Acknowledgements** We thank M. Albrechtsen for valuable discussions. We acknowledge financial support from the Villum Foundation Young Investigator Programme (grant no. 13170), Innovation Fund Denmark (grant no. 0175-00022—NEXUS and grant no. 2054-00008—SCALE), the Danish National Research Foundation (grant no. DNRF147—NanoPhoton), Independent Research Fund Denmark (grant no. 0135-00315—VAFL), the European Research Council (grant. no. 101045396—SPOTLIGHT), the European Union's Horizon 2021 research and innovation programme under the Marie Skłodowska-Curie Action (grant no. 101067606—TOPEX) and the European Union's Horizon research and innovation programme (grant no. 101098961—NEUROPIC).

**Author contributions** A.N.B. and T.A.S.W. fabricated the devices and performed the SEM characterization. A.N.B. and G.A. performed the optical characterization. B.V.L., A.N.B., G.A. and K.T. carried out the numerical design and analysis. S.K. performed the STEM measurements. A.N.B., T.A.S.W. and G.A. carried out the data analysis. G.A., A.N.B. and S.S. prepared the manuscript with input from all authors. B.V.L., K.T., A.N.B., T.A.S.W. and S.S. designed the experiment. S.S. conceived, initiated and supervised the project with co-supervision by B.V.L. and G.A.

**Competing interests** S.S. is a co-founder and shareholder of Beamfox Technologies ApS, which provided the software Beamfox Proximity that was used for proximity-effect correction.

**Additional information**
**Correspondence and requests for materials** should be addressed to Ali Nawaz Babar or Søren Stobbe.
