## [Peer Review File · Nature]

Manuscript Title: Self-assembled photonic cavities with atomic-scale confinement

Reviewer Comments & Author Rebuttals

Redactions – unpublished data

Reviewer Reports on the Initial Version:

Referees' comments:

Referee #1 (Remarks to the Author):

The work reported in the manuscript by Babar et al. is of accurate technical quality, and it communicates novel and relevant results in the nanotechnology domain, specifically in the quest for the realization of unprecedented figures of merit for nanostructured devices in Photonics. Even if a few details might be improved (see comments below), the manuscript is well written, and the presentation of results is of high quality overall, with more than 20 pages of additional clarifying data and results reported in the supplementary information file.

Briefly, the report presents a thorough procedure based on conventional electron beam lithography and etching, but still allowing to produce void features at the nanometer level by exploiting the balance between surface attracting forces and elastic ones, which the authors define 'self-assembly' of photonic cavities. The self-assembly procedure is applied to the realization of sub-diffraction limit and high Q photonic resonators. Indeed, the electromagnetic field can be confined in such void features, after careful cavity design, experiencing an ultra-sub diffraction level confinement. Quantitatively speaking, the authors are able to show confinement in void features with effective single-mode volumes about two orders of magnitude lower than the cubed wavelength in air, with photon lifetimes in the order of 30 ps at telecom wavelengths (i.e., corresponding to a Q-factor of about 40k for the best structure characterized in the manuscript). Most importantly, it is shown that the self-assembly procedure can be adapted to realize integrated and suspended photonic resonators, specifically designed as 1D nanobeam photonic crystal cavities with bow-tie shaped holes, in- and out-coupled through tunnel-coupled waveguides whose access ports are self-assembled with the very same technique employed to deterministically adjust the nanocavity voids. The characterization of such devices is remarkable, showing very good figures of merit in terms of loaded Q-factor and transmission efficiency, which holds great promise in view of coupling single emitters to ultra-high Q/V ratio modes.

In view of the results achieved, and the potential impact it could have on biosensing, Raman spectroscopy, nonlinear optics, the work is at the level that would be suited for a Nature publication, for which I recommend the Editors to consider it. However, there are a number of details in the presentation, or questions that arose while reading the manuscript and its supplementary material, which should be considered before further action be taken, in my opinion. Here is a brief summary of the points to be addressed, not necessarily in order of importance or relevance.

- The manuscript is well conceived, overall. However, the impression is that the same story could have been told in less pages in the main text, while shifting some of the information to the SI file. In fact, the latter is already made of about 24-25 pages, and it is quite essential to understand some of the results presented in the main manuscript (as also witnessed by the constant recall to Sections and Figures from the main text). Personally, I did not understand at all what exactly the offset, δ , was meaning, before reading carefully the SI. Also, Figure 1 is quite explanatory, but panel 1d is really awkward to read, I had to look very carefully to understand what information can be extracted from there. I am sure the authors can find a better way of presenting these data, although I have not a clever suggestion myself, at the moment. The whole procedure leading to the build up of the self-assembly protocol, pages 3-5, could be significantly shortened, and some

of the panels (despite being quite nice and significant SEM or STEM pictures) could go to Supplementary file. Finally, concerning the presentation of the resonators results, I also have the feeling it could be compacted somehow. For example, the authors present measurements on samples with the aim of characterizing the intrinsic Q-factor of the cavities, and later measurements on different samples made for integration with input/output ports. However, an alternative idea could be to measure resonant scattering from the same samples in which transmission measurements are collected, and cross-check the resonances with the ones found in transmission. All the set of measurements presented in Fig. 4, for example, could be shifted to the Supplementary file.

What I mean is, essentially, that this work might be presented as a Letter, without the need to go for a full 10-pages article, also considering that there are already 25 pages of essential information left aside, and adding some more would not bother.

- The resonant scattering measurements are relevant to characterize intrinsic Q-factors, but they require extreme sensitivity and might be affected by fitting errors from Fano lineshapes; are the error bars in Fig. 4 d,e,f derived from these errors, or they simply represent the spread of the measured values? A more quantitative discussion on the fitting should be provided, also in the supplementary file, even if this is now quite standard procedure in nanophotonics.
- The scattering induced losses might be extremely relevant when the field intensity is peaked in nm-scale void features, but the measurements indicate that the experimental Q-factors are reduced only by a factor of 2-3 as compared to the theoretical ones. One question that naturally arises is whether the authors have performed a modelling analysis of the losses induced by surface roughness on such tightly confined modes. If not, I believe they have all the tools at hand (Finite-element software COMSOL) to possibly perform such an analysis, and thus confirm that the order of magnitude in Q-factor reduction is actually consistent with the experimental outcomes.
- Measurements in transmission are described in more detail in the SI file: the authors employ a cross-polarized confocal setup allowing to independently in- and out-couple from the access gratings; transmission measurements reported in the manuscript are normalized to the reference transmission taken from a suspended waveguide having the same width of the nanobeam resonators, i.e., without holes. This procedure is formally correct, but it does not give enough quantitative information: it would be useful, for completeness, to also present, maybe in the SI, the raw data showing the absolute transmission, maybe also the reference waveguide. In particular, looking at Fig. S16, the sidewall roughness of the suspended waveguide bridge seems quite consistent, which might indicate that the absolute transmission of the whole device might be reduced to very low levels in this samples.
- The discussion regarding the mode volume should be improved, in my opinion. This figure of merit is presented and discussed in several parts of the manuscript, but a formal definition is never given, not only in the main text but also in the SI file. Given the relevance of the result, and the fact that, among all the figures of merits presented in this work, the mode volume is the one that cannot be directly accessed through measurements but only inferred from numerical simulations (whose accuracy should be discussed, by the way, see point below), the mode volume definition should be explicitly reported. Some of the authors have already reported experimental and theoretical results on different type of resonators (e.g., Refs. 5 and 29 in the manuscript), but these should be reported here as well, for self-consistency.
- In the transmission spectra reported in Fig. 5f, and also in S17, an explicit comparison between a mirror and its corresponding nanobeam resonator is shown. However, in all these spectra the long wavelength edge of the mirror is not present, at least in the wavelength range shown in these figures (neither in Fig. 5f, nor in the corresponding spectra reported in the SI for different, nominally identical, devices in Fig. S17), while it is clearly evident in all the nanobeam spectra to fall around 1550 nm. Is there a reason for that?

- In general, there is a lack of quantitative information regarding the numerical simulations employed to either design the bowtie cavities or compare with experimental results. From the SI file we realize that all the simulations reported in the manuscript are based on the Finite-element software COMSOL. In my opinion, at least in the SI file it would be useful to also include some details such as resolution, simulation time, convergence tests (if any).
- The resonant wavelength increases on decreasing the air gap within the bowtie tips, as clearly shown in Fig. 4d. I am sure this is correct, as it is also reproduced from COMSOL simulations, however, can the authors give an argument for why this is the case? In fact, intuitively one might think that reducing the air gap, and decreasing the mode volume, corresponds to increasing the field intensity in the void (i.e., the lower refractive index) region, thus reducing the effective mode index, which should actually lead to a blue-shift of the mode...I am sure this argument does not apply, but I cannot figure out where I'm wrong here. Just a comment might be useful.
- How can the spring constant of each structure be actually measured? the results are reported as a function of k/A , which is the relevant parameter, e.g., in Eq. S4; but which Area are the authors assuming to normalize the elastic constant? Or are they actually determining k/A directly? And in which way?

Referee #2 (Remarks to the Author):

This manuscript proposes and demonstrates a technique to form nm-wide, high aspect ratio gaps and voids at the intended locations, which cannot be formed by direct patterning. This technique utilizes the pulling force generated at the gap to close the gap after the structure is released from the underneath SiO₂ sacrificial layer, where proper patterns are formed in the gap region in order to leave a small gap or void. Using this technique, they have formed an optical cavity with a bowtie structure that has a gap width of 1~3 nm and a gap height of 220 nm, and achieved a measured Q value of 1.5×10^4 and an estimated mode volume of 8.8×10^{-4} . This is highly evaluated as a new method to fabricate nm-level gaps and voids with good reproducibility.

On the other hand, self-assembly methods using appropriate patterning and force after release have been conventionally utilized in the field of MEMS/NEMS. (For example; J. of Microelectromechanical Systems 12, 387-417 (2003), small 5, 1600-1630 (2009), and Nano Lett. 21, 3205-3210, (2021).) Although the scale and usage may be different from the one in this report, the basic concept of the method is not largely different from those in the field of MEMS/NEMS. Therefore, I think that the novelty of the technology in this report compared to those of the conventional MEMS/NEMS should be discussed to clarify the difference. Also, I think the claim in the title and introduction is a bit too large, since the technology in this report is basically to create gaps and voids at the nm level, i.e. a claim like "self-assemble of atomic-scale gap" may be appropriate.

However, the fact that a gap of about 1 nm was formed with high fabrication accuracy is highly commendable. In this regard, it is thought that an extremely vertical etching profile is required to apply this method. Therefore, I think that the relationship between verticality and the gap formed should be discussed. In addition, it is necessary to discuss whether a special method is necessary for achieving such high verticality, as well as the reproducibility and pattern dependence of the verticality.

There are also several questions regarding the optical cavity part. First, in Nat Commun. 13, 6281 (2022), the same group reported that a mode volume of 3×10^{-4} and a measured Q value of 1000 can be achieved without nanogaps and with a very complicated structure. We would like to know the relationship between that structure and the structure in this manuscript. By the way, in Fig. 5, the delta is 12 nm, but in accordance with the description of L347-348, it seems that the gap is closed in this case. Also, in Fig. 4f, the mode volume is the smallest when the gap is closed.

If so, to what extent does this technology, which can reproducibly create a nanogap, contribute to the realization of a very small mode volume? Perhaps it is easier to create a gap of $0\sim 1$ nm using the method in this report than to create a continuous bridge of 2 nm width in the previous report. Also, I think the method and structure in this report is better in terms of the possibility of introducing a light emitter, etc. However, I would like to see a discussion of this issue. Another excellent demonstration in this manuscript is to make the resonator part and the coupling waveguide part separately and attach them together by self-assembly.

Considering all the above, I think that the value of the results in this manuscript is in the fact that it was actually possible to create a very fine structure with good reproducibility, and the fact that a bowtie cavity with a small gap and good optical characteristics was fabricated using this method. Although it is difficult to judge, I am afraid that this manuscript at present form is more suitable for more specialized journals such as Nature Nanotechnology and Nature Photonics than Nature.

Referee #3 (Remarks to the Author):

Babar et al. describe their work on "self-assembly of atomic-scale photonic cavities" using van der Waals attraction and pull-in instability. The manuscript is well-written and presents an intriguing concept for the nanofabrication of photonic devices made of underetched silicon. The authors even present the far-field cross-polarized optical characteristics of these cavities and their integration with photonic circuitry. However, I have some concerns about the technical points presented, which prevent me from recommending this work for publication in Nature at this time.

1) Firstly, the most critical point that needs to be addressed is the resolution of the final gap in the bowtie regime. While it is clear that surface roughness can determine the gap for the case of van der Waals contact of flat surfaces (as demonstrated in Fig. 1), for the case of the bowtie nanocavity, the gap should not be determined by surface roughness (r) as the authors suggest, but by the offset (δ) between the flat region and the bowtie region during the collapsing of the two nanofabricated halves. This offset, in turn, is determined by conventional nanofabricating tools such as electron beam lithography and reactive-ion etching used in this study, which could lead to substantial uncertainty in the fabrication of large-scale devices, impacting scalability and reproducibility.

To explain this further, imagine taking two plates that are both rough and nanopatterned, and then collapsing them. In the approximation of hard bodies, the geometrical constraints do not allow for a higher precision gap region than the offset (δ). Although the native SiO₂ formed at the surface may vary in thickness and roughness, it will be the same in both the flat and bowtie regions. It is unclear how the resolution of the final gap region can exceed the standard resolution of nanolithography used to produce the halves. Therefore, the authors should provide convincing arguments or a cartoon to explain this, or else adapt their message accordingly. While the authors may have been able to "fish out" bowties with sub-nanometer gaps in thousands of fabricated devices, it is unrealistic to claim that this can be scaled up massively, as the introduction and abstract seem to claim (the uncertainty in the fabrication of large-scale bowtie devices will be still determined by standard nanofabrication).

2) Secondly, questions arise about the quality and precision of vertical etching. In particular, it is unclear how vertical the etching is for a 200 nm thick Si and how this impacts the final device.

There are also some minor points that should be addressed, such as:

- specifying the color map scale in Fig. 2b
- adding a reference to Hu et al., Sci. Adv. 2018;4:eaat2355
- improving references to self-assembly and its combination with top-down methods, as in the current version they are relatively outdated.

Finally, based on the above concerns, it is unclear how the proposed method can lead to atomic-scale photonic cavities, and the current title seems like an oversell. Instead, it seems like the method has the potential to produce nanometer-to-subnanometer photonic cavities, with the accuracy of the gap still determined by conventional nanolithography steps instead of surface roughness. Therefore, I suggest the authors revise their title and replace "atomic-scale" with "subnanometer" to better reflect their findings. Also, the authors should explicitly clarify the feasibility of a massive scale-up of these bowties in the sub-nanometer regime, or else explain how this resolution is reached and provide experimental evidence for such precision and accuracy on a large scale.

Author Rebuttals to Initial Comments:

Response to the Reviewers' comments on our manuscript:

Self-assembled photonic cavities with atomic-scale confinement

Contents

Note for the Reviewers	1
Reviewer #1	2
Reviewer #2	28
Reviewer #3	42
References	50

NOTE FOR THE REVIEWERS

We thank all three Reviewers for their careful and constructive feedback and criticism on our manuscript. In the following, we provide verbatim copies of the Reviewers' remarks in bold types followed by our response in normal font. We hope that our response resolves all questions and comments.

REVIEWER #1

The work reported in the manuscript by Babar et al. is of accurate technical quality, and it communicates novel and relevant results in the nanotechnology domain, specifically in the quest for the realization of unprecedented figures of merit for nanostructured devices in Photonics. Even if a few details might be improved (see comments below), the manuscript is well written, and the presentation of results is of high quality overall, with more than 20 pages of additional clarifying data and results reported in the Supplementary Information.

Briefly, the report presents a thorough procedure based on conventional electron beam lithography and etching, but still allowing to produce void features at the nanometer level by exploiting the balance between surface attracting forces and elastic ones, which the authors define 'self-assembly' of photonic cavities. The self-assembly procedure is applied to the realization of sub-diffraction limit and high Q photonic resonators. Indeed, the electromagnetic field can be confined in such void features, after careful cavity design, experiencing an ultra-sub diffraction level confinement. Quantitatively speaking, the authors are able to show confinement in void features with effective single-mode volumes about two orders of magnitude lower than the cubed wavelength in air, with photon lifetimes in the order of 30 ps at telecom wavelengths (i.e., corresponding to a Q-factor of about 40k for the best structure characterized in the manuscript). Most importantly, it is shown that the self-assembly procedure can be adapted to realize integrated and suspended photonic resonators, specifically designed as 1D nanobeam photonic crystal cavities with bow-tie shaped holes, in- and out-coupled through tunnel-coupled waveguides whose access ports are self-assembled with the very same technique employed to adjust the nanocavity voids deterministically. The characterization of such devices is remarkable, showing very good figures of merit in terms of loaded Q-factor and transmission efficiency, which holds great promise in view of coupling single emitters to ultra-high Q/V ratio modes. In view of the results achieved, and the potential impact it could have on biosensing, Raman spectroscopy, nonlinear optics, the work is at the level that would be suited for a Nature publication, for which I recommend the Editors to consider it. However, there are a number of

details in the presentation, or questions that arose while reading the manuscript and its supplementary material, which should be considered before further action be taken, in my opinion. Here is a brief summary of the points to be addressed, not necessarily in order of importance or relevance.

We thank the Reviewer for the careful review and valuable feedback, and we are pleased that the Reviewer “recommend[s] the Editors to consider” our manuscript for publication in Nature contingent on us addressing a number of points, which we have carefully addressed in the revised manuscript. Our detailed response to the specific points follows below.

- The manuscript is well conceived, overall. However, the impression is that the same story could have been told in less pages in the main text, while shifting some of the information to the Supplementary Information. In fact, the latter is already made of about 24-25 pages, and it is quite essential to understand some of the results presented in the main manuscript (as also witnessed by the constant recall to Sections and Figures from the main text).

We aimed to keep the main text coherent and readable even without consulting with the Supplementary Information (SI), but we realize from the comments raised by all three Reviewers that our initial submission had some shortcomings in its presentation, including too many recalls to the SI. While our response to the reviews has required us to add a number of details to the revised manuscript, we have rearranged the figures with the net effect that the revised manuscript has four rather than five figures, and the text has been shortened following the Reviewer’s advice. In addition, to increase the readability of the main text, we have now grouped the recalls into Sections in the SI to the end of the different paragraphs whenever possible.

Personally, I did not understand at all what exactly the offset, δ , was meaning, before reading carefully the SI.

We thank the reviewer for bringing up this point, which made us realize that the explanation of the offset, δ , in the previous version of the manuscript was too brief. In the revised manuscript, we have rearranged Fig. 2d to make room for explaining the offset explicitly already at an early stage of the manuscript (For completeness, the revised Fig. 2 is reproduced here along with the revised caption as Fig. R1d). This has also allowed us to briefly highlight the offset’s meaning in the caption. We hope that the revisions to the figure, its caption, and the associated description in the main text

Response Figure R1: [FIGURE 2 OF THE REVISED MANUSCRIPT] **Design and fabrication of self-assembled silicon nanobeam bowtie cavities.** a, The geometry of the nanobeam bowtie cavity. b, Normalized electric field of the cavity mode, $|E|$, with a logarithmic color map. c, Normalized electric field, $|E|$, of the central bowtie unit cell of the nanobeam cavity, with a linear color map. d, Electron-beam-lithography mask schematic of the central region of the nanobeam cavity, with black (grey) illustrating exposed (unexposed) areas. The offset, δ , is defined as $\delta = (g_P - g_T)/2$. e, Tilted (20°) SEM image of the central part of the cavity before the release etch triggers the self-assembly of the two parts initially separated by $g_f = 50$ nm, except at the bowtie where the distance is g_b . f, Tilted (20°) SEM of the central part of a nanobeam cavity after self-assembly, with the approximately 2 nm gap indicated in the zoom-in. g, Top-view SEM image of the full device, including the spring suspension. h-i, Top-view high-resolution scanning transmission electron microscope (STEM) images of the central unit cell and their bowtie tips for cavities fabricated using $\delta = 10$ nm and $\delta = 11$ nm. The scale bars are calibrated using the 0.19 nm inter-planar distance of the visible (022) crystalline silicon planes.

make the role of δ clear without the need to refer to the details in the SI.

Also, Figure 1 is quite explanatory, but panel 1d is really awkward to read, I had to look very carefully to understand what information can be extracted from there. I am sure the authors can find a better way of presenting these data, although I have not a clever suggestion myself, at the moment.

Upon revisiting Fig. 1d with the Reviewer's comment in mind, we agree that including information about the widths of the different platforms arguably made it too complex. We maintain that this information is important for future studies of self-assembly by surface forces, so we have moved the previous version to the SI and replaced Fig. 1d with a simplified version (see Fig. R2 here), whose data points now only indicate if the device with the given initial gap and spring constant per area collapsed or not. We find this version much simpler to understand, and we hope the Reviewer agrees.

The whole procedure leading to the build-up of the self-assembly protocol, pages 3-5, could be significantly shortened, and some of the panels (despite being quite nice and significant SEM or STEM pictures) could go to the Supplementary file.

We agree that the number of SEM and STEM images in our manuscript is slightly unconventional, but we are confident that their use is essential for communicating our main results. The purpose is in part to visualize complex devices and more importantly for illustrating, explaining, and documenting the self-assembly protocol. For example, the SEM images are central to explain how the cavities and the surrounding circuits are fabricated, and the STEM images are central to document that our method allows fabricating atomic-scale features. Having said that, we have made significant shortenings of both text and figures in the revised submission as described in detail below.

Regarding the main text, we acknowledge that some explanations could be shortened and/or moved to the SI. In the revised manuscript, we have reduced the explanations as follows:

- The introduction has been shortened.
- The first paragraph in the "Deterministic self-assembly by surface forces" section has been shortened.
- The discussion of the STEM images has been shortened, although some additional aspects regarding sidewall verticality have been added following the comments by Reviewers 1 and 2.
- Figures 2 and 3 have been merged into a single figure and the resonant-scattering raw spectral data has been moved to the SI. (The revised Figs. 2 and 3 are reproduced here as Fig. R1 and

Response Figure R2: [FIGURE 1d OF THE REVISED MANUSCRIPT] Measured map of the design space for self-assembly with compliant silicon structures, obtained by characterizing 1536 platforms by SEM. The green-filled circles represent the collapsed platforms, and the purple-filled represent the non-collapsed platforms. All the devices below the upper bound collapse, while those above the lower bound do not.

Fig. R3).

Finally, concerning the presentation of the resonators results, I also have the feeling it could be compacted somehow. For example, the authors present measurements on samples with the aim of characterizing the intrinsic Q-factor of the cavities, and later measurements on different samples made for integration with input/output ports. However, an alternative idea could be to measure resonant scattering from the same samples in which transmission measurements are collected, and cross-check the resonances with the ones found in transmission. All the set of measurements presented in Fig. 4, for example, could be shifted to the Supplementary file.

First, we would like to stress that the two types of cavities in our study are – despite their strong geometric similarities – rather different precisely because they target those experimental measurement conditions. The cavities used for resonant scattering measurements have short tapering sections and long mirror sections, which minimizes in-plane losses but limits the achievable quality factor due to

Response Figure R3: [FIGURE 3 OF THE REVISED MANUSCRIPT] **Resonant scattering for self-assembled nanobeam cavities.** **a**, Representative cross-polarized scattering spectrum, R_{cross} (purple line), of a cavity fabricated with $\delta = 8$ nm. The fitted Fano lineshape over the fitting range is included (black line) and the extracted quality factor, Q , is given. **b**, Simulated and measured resonant wavelengths, λ_0 , quality factors, Q , and effective mode volumes, V_{eff} , as a function of the bowtie width, g . The simulations use all measured dimensions, including a 2 nm native oxide layer. The theory curve for λ_0 is red-shifted by 15 nm due to the variations in the device layer thickness and the SEM extracted dimensions. The experimental bowtie widths are estimated following the SI Section S4.3, and the error bars show the standard deviation obtained from measurements on a set of nominally identical cavities.

out-of-plane losses. Longer defects also minimize those and lead to overall larger Q s, but the emission pattern cannot be captured with our microscope objective, which has a limited NA of 0.42 (Mitutoyo Plan Apo NIR 50X). On the other hand, the cavities used for transmission measurements have a longer defect, which limits out-of-plane losses, but shorter mirror sections that allow a considerably high on-resonance transmission. We refer to the SI Section 3.1 for further details on the cavity designs.

Second, while we agree that the transmission data are particularly important, we also believe that it is important to include the resonant-scattering data set in the main text because the cavities

used for the resonant scattering measurements are the same as those shown in Fig. 2 of the revised manuscript, so this data set is important for correlating the structural characterization with the optical experiments. In addition, we are convinced that jumping directly to the integration of self-assembled cavities with large-scale circuitry without explaining the basic self-assembly of cavities would make the manuscript rather hard to read and follow.

What I mean is, essentially, that this work might be presented as a Letter, without the need to go for a full 10-pages article, also considering that there are already 25 pages of essential information left aside, and adding some more would not bother.

To summarize our efforts in shortening the revised manuscript, we followed the Reviewer's advice and we have merged two figures into the new Fig. 2 (see Fig. R1 here). We have also moved the raw resonant-scattering data to SI Section 5.1 and kept only the most essential parts in the revised Fig. 3, which is now also only single-column. In addition, we have substantially shortened the main text, but we maintain that the results presented in the main text of the revised manuscript are essential to understand our work, and we are therefore reluctant to make further reductions. We find the revised manuscript more concise and we hope that the Reviewer agrees. Following the advice from the Reviewer as well as from the two other Reviewers, we have added a substantial amount of material (in total approx. 30 pages including new simulations and figures) to the SI in the revised submission. Finally, we would like to note that since 2020, Nature no longer publishes Letters, so in that sense, the manuscript would, if published, in any case, be in the form of an Article.

- The resonant scattering measurements are relevant to characterize intrinsic Q-factors, but they require extreme sensitivity and might be affected by fitting errors from Fano lineshapes; are the error bars in Fig. 4 d, e, f derived from these errors, or they simply represent the spread of the measured values? A more quantitative discussion on the fitting should be provided, also in the supplementary file, even if this is now quite standard procedure in nanophotonics.

We thank the Reviewer for commenting on the errors coming from the fits, which were not considered in the previous version of the manuscript. The error bars in Fig. 4d and e represented the standard deviations of the measured (fitted) values across the fundamental modes of a number of cavities with nominally-identical lithography mask. These results are now shown in Fig. 3b in the revised manuscript, and we refer to the numbers in the revised manuscript in the following. The reason why we choose to use that for the error bars is that such spread is much larger than the uncertainty coming

from the fit, as we explain below.

We fit our cavity spectra around the observed resonant features using a Fano lineshape,

$$F(\omega) = A_0 + F_0 \frac{[q + 2(\omega - \omega_0)/\Gamma]^2}{1 + [2(\omega - \omega_0)/\Gamma]^2}, \quad (\text{R1})$$

where ω is the frequency, ω_0 is the resonance frequency of the cavity mode, Γ its linewidth, and q measures the relative amplitudes between the main and the background modes. We note that q can have both positive and negative values and determines the lineshape asymmetry. A_0 and F_0 are constant scaling factors. We use a nonlinear regression to perform the fit. The fitted lineshape, corresponding fit parameters, the coefficient of determination R^2 , and the extracted quality factor including its uncertainty are shown in Figs. R4, R5, and R6 for δ values of 8 nm, 9 nm, and 10 nm, respectively. The given uncertainty in Q , σ , is computed by propagating the errors of the resonance wavelength and the linewidth, which are obtained as the square root of the diagonal elements of the regression covariance matrix. In particular, we specify the uncertainty in Q as $\pm 2\sigma$ (95% confidence interval).

In summary, while we acknowledge that it is harder to obtain good fits to Fano lineshapes compared to Lorentzians, the R^2 values range between 0.95 and 1 across the entire data set and the resulting uncertainties in Q are well below the spread in the values, so we have kept the error bars as in the initial submission. Nevertheless, we acknowledge that the details reported here are valuable and that individual Q -factors should be reported with their own uncertainty, so these details and associated figures are now included in the SI Section 5.1 and the single-cavity Q s are reported with their uncertainty.

Response Figure R4: Fano lineshape and fitted parameters of cavity resonances for self-assembled cavities with $\delta = 8$.

Response Figure R5: Fano lineshape and fitted parameters of cavity resonances for self-assembled cavities with $\delta = 9$.

Response Figure R6: Fano lineshape and fitted parameters of cavity resonances for self-assembled cavities with $\delta = 10$.

- The scattering-induced losses might be extremely relevant when the field intensity is peaked in nm-scale void features, but the measurements indicate that the experimental Q-factors are reduced only by a factor of 2-3 as compared to the theoretical ones. One question that naturally arises is whether the authors have performed a modelling analysis of the losses induced by surface roughness on such tightly confined modes. If not, I believe they have all the tools at hand (Finite-element software COMSOL) to possibly perform such an analysis, and thus confirm that the order of magnitude in Q-factor reduction is actually consistent with the experimental outcomes.

The question of roughness-induced scattering losses in the self-assembled bowtie cavities is very relevant. The reason is two-fold. First, and as the Reviewer suggests, the field confinement in nm-scale void features makes the cavities extremely sensitive to sidewall roughness (see, e.g., our recent publication [1]). Second, the interface between the two self-assembled halves is not perfect as it originates from two surfaces with a small yet unavoidable level of sidewall roughness before self-assembly.

The Reviewer comments that the experimental Q-factors are reduced only by a factor of 2-3 as compared to the theoretical values, and we agree that this relative change would be small if it had pertained to experiments on conventional nanobeam cavities with ultra-high Q. However, we would argue that such a drop is actually quite large in our case, as explained in the following. The experimental value of Q is governed not only by disorder-induced extrinsic scattering, Q_{ext} , but also by the radiation-limited intrinsic scattering, Q_{int} . This is often written, for large enough disorder, in a Mathiessen-like rule as [2]

$$\frac{1}{Q} = \frac{1}{Q_{\text{int}}} + \frac{1}{Q_{\text{ext}}}. \quad (\text{R2})$$

The key aspect here is that, for the cavities we have designed, Q_{int} is not very large (We design the cavities to have Q_{int} in the range $(4 - 10) \cdot 10^4$), at least compared to typical ultra-high-Q nanobeam cavity designs with Q_{int} of several million. Therefore, what might appear as a small drop is actually not a sign of the cavities having low scattering losses, but rather a sign that they have low Q_{int} . Actually, we have indications of considerable scattering losses due to the tight field confinement: We have recently performed measurements on nanobeam photonic-crystal cavities (with a design similar to that in Ref. [3]) where we observe good agreement of Q between theory and simulation up to Qs around $4 \cdot 10^5$, indicating indeed that Q_{ext} for the bowtie cavities is much smaller than for other cavity designs, i.e., they are more sensitive to structural defects. On a related note, we have recently reported in Ref. [4] record-low propagation losses in W1 photonic-crystal waveguides fabricated using the same lithography and etching used for the bowtie cavities here, which evidences that our top-down nanofabrication is state of the art. The bowtie cavities we explore can in principle be made to have

much larger Q_{int} by making longer defect regions and using more mirror unit cells. If we had done that, in which case the possibility of measuring them would be compromised (due to the low NA of our objective for resonant scattering and the strongly reduced transmissivity for our transmission measurements, as discussed already above), the drop factor would have been very large.

The reviewer also suggests confirming (in order of magnitude) the observed drop by carrying out full-scale finite-element COMSOL simulations of bowtie cavities with geometries that include realistic disorder. We agree that such simulations would be of great importance, but they are, unfortunately, not entirely realistic at this point. There are two reasons for this: The first is that modelling disorder is generally hard because it requires detailed knowledge of the statistical properties (amplitude and disorder correlation) for all kinds of disorder (sidewall roughness, surface roughness, line-edge roughness, sidewall angle fluctuations, chemical contamination from etching, etc.). For this reason, realistic treatments of disorder in nanophotonics constitute a major outstanding challenge, which has not been resolved in the literature so far. The typical approach in the literature is to assume that all kinds of disorder can be projected onto a single kind of disorder, but even so, the lack of knowledge about the correlation length of the disorder makes this a highly non-trivial task. The second reason, which is particular to the cavities studied in our work, is that the quasi-normal-mode (QNM) calculations that we run to obtain converged figures of merit (mode volume, quality factor, and resonance wavelength) take approximately 1.5 hours per cavity and use 150 GB of RAM on an AMD EPYC 7702P 64-Core server (see point below and SI Section 3.3 on numerical simulations and mesh convergence). The finite-element meshes would need to be scaled down dramatically in size to model surface roughness at a scale below a nanometer. The full-scale simulation of even a single cavity realization, including realistic disorder, is, therefore, a computationally intractable problem with our resources. We have, of course, access to cluster computing resources with many more cores and much more memory, but COMSOL scales very poorly on shared-memory hardware.

Despite the intractability of full-scale realistic disorder, we have explored the role of particular types of disorder to elucidate what may (and may not) be contributing to the drop in Q . In our structures, we identify line-edge roughness (see Fig. S31 in the SI) and bowtie width fluctuations (see Fig. S21 in the SI) as the dominant types of disorder. First, we have used a QNM non-Hermitian first-order perturbation-theory approach [5] to model the effect of line-edge roughness, which is, by construction, numerically tractable since it only requires the QNM calculation of the unperturbed cavity. We consider the effect of shifting the boundaries between two materials (labeled as 1 and 2), following a function $h(\mathbf{r})$ directed along the normal between the two materials and pointing from material 1 to material 2. The first-order complex shift to the complex eigenfrequency $\tilde{\omega}_n$ of a QNM is

Response Figure R7: The effect of line-edge roughness: a quasi-normal-mode first-order perturbation theory approach. **a**, Half-bowtie geometry indicating in blue the surfaces where line-edge roughness is considered for each bowtie. **b** The two functions entering Eq. R3 for the central bowtie: (top) Real and imaginary part of $F(s)$ and (bottom) single realization of the normally-oriented line-edge roughness profile, $h(s)$, on the top and bottom sidewalls of the bowtie (as shown in **a**), taken from an exponentially-correlated ($L_c = 10$ nm) normal distribution $\mathcal{N}(\mu = 0, \sigma = 0.75$ nm). **c** Top-view visualization of the bowtie centre including the roughness profiles. **d** Histogram of the resonant wavelengths obtained from 10^5 roughness realizations. **e** Same as **d** for the quality factor, Q . The vertical lines in the histograms indicate the values in the unperturbed case.

given by

$$\begin{aligned} \Delta\tilde{\omega}_n &= -\frac{\tilde{\omega}_n}{2} \int_S [(\epsilon_1 - \epsilon_2) \mathbf{E}_n^{\parallel}(\mathbf{r}) \cdot \mathbf{E}_n^{\parallel}(\mathbf{r}) - (\epsilon_1^{-1} - \epsilon_2^{-1}) \mathbf{D}_n^{\perp}(\mathbf{r}) \cdot \mathbf{D}_n^{\perp}(\mathbf{r})] h(\mathbf{r}) dS \\ &= \int_S F(s) \cdot h(s) dS \end{aligned} \quad (\text{R3})$$

where \mathbf{E}_n and \mathbf{D}_n are the normalized complex electric and displacement fields of the QNM, the

Response Figure R8: The effect of bowtie width and position fluctuations: full-scale simulations
 a, Bowtie tip geometry, indicating the bowtie width, g , which randomly departs from its average value of 2 nm, according to a uniform distribution, $\mathcal{U}(-1, 1)$. **b** Histogram of the resonant wavelengths, Q -factors and mode volumes, V , obtained from 100 cavity realizations. The vertical lines in the histograms indicate the values obtained for the unperturbed case. **c,d** Same as **a** and **b** for the positional disorder of the triangles forming the bowtie, with disorder considered as a departure from the nominal positions by a quantity determined from a normal distribution, $\mathcal{N}(0, 1)$.

superscripts “ \parallel ” and “ \perp ” denote field components respectively parallel and perpendicular to the shifted boundary, S . Figure R7 summarizes the type of results we obtain for a nanobeam bowtie cavity with average bowtie width $g = 2$ nm, where, for simplicity, we have omitted the oxide layer. Due to the tightly confined fields, we consider only line-edge roughness at the bowtie tips, specifically over the surfaces indicated in blue in Fig. R7a, defined by the intersection of the bowtie boundaries with a small circle of radius $R = 20$ nm. Both the real and the imaginary part of the function $F(s)$ in Eq. R3 fall off rapidly away from the bowtie tips (see Fig. R7b for the central bowtie), and it can therefore be assumed that only structural disorder in the small spatial regions near the bowtie tips must be included to capture the full first-order effect of roughness. Figure R7b also includes two realizations of the line-edge roughness profiles, $h(s)$, taken from an exponentially correlated ($l_c = 10$ nm) normal distribution, $\mathcal{N}(\mu = 0, \sigma = 0.75$ nm). Note that we have convoluted the random profile with a Gaussian kernel ($\sigma_f = 1$ nm, black line) to avoid unrealistically sharp geometries (grey line) and that the values of the correlation length, l_c , and standard deviation σ are chosen for illustrative purposes and not based on careful structural characterization. Such profiles are created for each bowtie in the cavity (see Fig. R7c for an example of a rough bowtie) and Eq. R3 is used to compute the resonant wavelength and Q -factor of the perturbed cavity. We generate profiles for 10^5 cavities and Fig. R7d and e show the

obtained histograms for both parameters. We observe well-behaved probability distribution functions, with normal-distributed wavelengths and log-normal distributed Q -factors. However, we see that the distribution for Q , e.g., its mean, does not drop compared to the Q of the unperturbed cavity, indicated with a vertical line. This stems from the fact that first-order perturbation theory is, by construction, unable to capture symmetry-breaking, which is inherent to the actual roughness and the main reason behind the shift of the distribution towards lower Q [6].

The second type of disorder we have explored is fluctuations in the bowtie width (Fig. R8a). In this case, full-scale simulations are at reach despite the added computational charge that a quarter instead of an eighth of the nanobeam needs to be solved for. Figure R8b shows the histogram of the resonant wavelength, Q -factor, and mode volume obtained from a set of 100 cavity realizations that include bowtie-to-bowtie width uniformly-distributed, $\mathcal{U}(-1, 1)$, fluctuations. We see that bowtie width fluctuations do not undermine the quality factor, but, on average, increase it. This may seem surprising, but it stems from the very limited length of the cavity defect region which makes a smooth potential impossible to realize. We have confirmed that longer cavities lead to a drop in Q , but understanding the crossover between the two cases would require a much more extensive simulation study that is beyond the scope of this review. In addition, the large net blue-shift we observe has previously been noticed and explained via perturbation theory and predicted to scale as the inverse of the mode volume [7], producing a shift of several nanometers in our case. In any case, what these simulations reveal is that the drop in Q we observe is not related to bowtie width fluctuations. To actually observe a considerable drop in Q , one needs to have full symmetry-breaking. We perform full-scale simulations (of half of the nanobeam, with the only remaining symmetry being that across the mid-plane of the slab) including positional disorder of the polygon forming the bowtie (Fig. R8c). This type of disorder breaks all the symmetries and therefore the full-scale simulations evidence a 37 % reduction of the mean Q relative to the unperturbed one for normally-distributed xy -position fluctuations, $\mathcal{N}(0, 1)$, as shown in Fig. R8d. Although our cavities were designed to use a fixed integer shot pitch, which fits the 1 nm electron-beam-lithography exposure grid and therefore minimizes centroid disorder in the lithography, fluctuations during development and etching will in general break the lateral mirror symmetries, which is also evident from, e.g., the STEM images shown in Fig. 2i of the main text.

The main outcome of the simulations presented here is the identification of breaking all the in-plane symmetries as a key element to account for the drop we observe in Q . Given that neither the perturbation theory study of line-edge roughness nor the full-scale simulations including bowtie width fluctuations include that key feature, we have decided to not include them in the updated Supplementary Information. Nevertheless, a discussion on the role of disorder and the full-scale simulations including positional disorder are now included in SI Section S3.5.

- Measurements in transmission are described in more detail in the SI file: the authors employ a cross-polarized confocal setup allowing to independently in- and out-couple from the access gratings; transmission measurements reported in the manuscript are normalized to the reference transmission taken from a suspended waveguide having the same width of the nanobeam resonators, i.e., without holes. This procedure is formally correct, but it does not give enough quantitative information: it would be useful, for completeness, to also present, maybe in the SI, the raw data showing the absolute transmission, maybe also the reference waveguide. In particular, looking at Fig. S16, the sidewall roughness of the suspended waveguide bridge seems quite consistent, which might indicate that the absolute transmission of the whole device might be reduced to very low levels in these samples.

Response Figure R9: Raw transmitted power spectra of relevant devices. Transmitted power of self-assembled nanobeam waveguide, bowtie mirror and bowtie nanobeam resonator, and conventional nanobeam waveguide for a laser power of $P = 10$ mW. All four devices use the same circuit crossings and have equivalent lengths.

We acknowledge that showing the raw spectra, including that used for normalization, is valuable to have quantitative information on the absolute power levels expected at a given incident power. However, we note that the designed circuits did not target a very efficient end-to-end transmittance since we have used a rather primitive one-ring grating coupler [8], which has a maximum theoretical

coupling efficiency of only 3 % and rather large in-plane reflectivity, but is extremely broadband and comes with rather relaxed alignment tolerances and therefore allows efficient circuit testing. Therefore, the low absolute transmittance levels that we report in Fig. R9 here and in the revised SI Section 5.2 are limited by the in- and out-coupling efficiencies of the grating couplers, and can be readily improved by using better coupler designs (see Ref. [9] for our recent publication on a much higher-efficiency grating coupler). Figure R9 includes the raw transmission spectrum (thin yellow line) and smoothed transmission spectrum (thick yellow line) of the self-assembled waveguide, the latter used to normalize the spectra in Fig. 4 in the main text. In addition, Fig. R9 includes the spectrum of a conventional waveguide of the exact same geometry (red lines). We observe a 25 % average drop in the waveguide transmission due to the self-assembly process, which stems from scattering losses due to the collapsed interface, although the possibility that the self-assembled and conventional waveguides bow differently (see for an example of such bowing Fig. 4c in the main text) cannot be ruled out.

- The discussion regarding the mode volume should be improved, in my opinion. This figure of merit is presented and discussed in several parts of the manuscript, but a formal definition is never given, not only in the main text but also in the SI file. Given the relevance of the result, and the fact that, among all the figures of merits presented in this work, the mode volume is the one that cannot be directly accessed through measurements but only inferred from numerical simulations (whose accuracy should be discussed, by the way, see point below), the mode volume definition should be explicitly reported. Some of the authors have already reported experimental and theoretical results on different types of resonators (e.g., Refs. 5 and 29 in the manuscript), but these should be reported here as well, for self-consistency.

In our work, the effective mode volume, V , of the explored cavities is evaluated by using the QNM,

$$\frac{1}{V} = \text{Re} \left[\frac{\epsilon_r(\mathbf{r}_0) \mathbf{E}(\mathbf{r}_0) \cdot \mathbf{E}(\mathbf{r}_0)}{\int_{V_T} \epsilon_r(\mathbf{r}) \mathbf{E}(\mathbf{r}) \cdot \mathbf{E}(\mathbf{r}) dV + i \frac{c\sqrt{\epsilon_r}}{2\omega} \int_S \mathbf{E}(\mathbf{r}) \cdot \mathbf{E}(\mathbf{r}) dA} \right] \quad (\text{R4})$$

where \mathbf{r}_0 is the position where mode volume is evaluated, $\mathbf{E}(\mathbf{r})$ is the electric field of the QNM, ϵ_r is the dielectric constant, the volume integral is taken over the entire simulation domain, V_T , and the surface integral is over the surfaces implementing the radiation boundary condition, S [10]. We select \mathbf{r}_0 to be the center of the nanocavity, which we define as the center of the central bowtie (at half the bowtie width and in the mid-plane of the silicon device layer). Evaluating at the center point is a robust method for calculating the mode volume because it is unaffected by the lightning-rod surface fields that appear at surfaces due to the electromagnetic boundary conditions [11, 12]. This definition

is now included in SI Section S3.2.

- In the transmission spectra reported in Fig. 5f, and also in S17, an explicit comparison between a mirror and its corresponding nanobeam resonator is shown. However, in all these spectra the long wavelength edge of the mirror is not present, at least in the wavelength range shown in these figures (neither in Fig. 5f nor in the corresponding spectra reported in the SI for different, nominally identical, devices in Fig. S17), while it is clearly evident in all the nanobeam spectra to fall around 1550 nm. Is there a reason for that?

The reason for that edge not being visible in the cavity spectra is simple: it falls beyond (or right at the edge) of the range of our tunable laser (Santec TSL-710). We refer to Fig. S9b in the SI to understand why the pass-band of the mirror does not match that of the corresponding nanobeam resonator. To build our cavity modes, we use the upper band in the band structure (often called the air band), and reduce its frequency by tapering from the mirror (black) into the defect (red) and back into the mirror, which builds a potential well. However, the same geometric tapering lowers the lower band (often called the dielectric band). This implies that in the long-wavelength region, i.e., the frequency region between the mirror and centre defect band edges, we cannot expect any transmission since, for these wavelengths, the central region is acting as a mirror. The distance between those two band edges corresponds to 83 nm, which is approximately the distance between the observed band edge for the mirror (around 1550 nm) and the edge of our laser (1640 nm). We have elaborated on these points in the revised SI.

- In general, there is a lack of quantitative information regarding the numerical simulations employed to either design the bowtie cavities or compare with experimental results. From the SI file we realize that all the simulations reported in the manuscript are based on the Finite-element software COMSOL. In my opinion, at least in the SI file it would be useful to also include some details such as resolution, simulation time, convergence tests (if any).

In our manuscript, we model nanobeam bowtie cavities using the finite-element method (FEM) and a complex eigensolver in COMSOL Multiphysics, which allow us to find the quasi-normal modes (QNMs) supported by the cavity. An illustration of the implemented numerical model is shown in Fig R10. We simulate one eighth of the nanocavity geometry (except for the disorder simulations discussed above) as allowed by the symmetry of the nanocavity geometry and of the fundamental mode

of interest. The symmetry of the latter is selected by setting the appropriate boundary conditions. In order to accurately capture the underlying physics, these calculations require extremely fine meshes in the bowtie region for single-nanometer bowtie widths. Therefore, simulating one eighth of the nanocavity structure is beneficial regarding required memory and computational power. In Fig R10, the blue region is the silicon, and the grey region is the air. Perfect magnetic conductor (PMC) boundary conditions are applied at the xy -plane and yz -plane. A perfect electric conductor (PEC) boundary condition is used at the xz -plane. The air box surrounding the nanobeam cavity has a height and depth of $L = 2.5 \mu\text{m}$. The length of the air box surrounding is equivalent to that of the nanobeam cavity. First-order scattering boundary conditions are used on the air box surrounding to model the radiation boundary condition at infinity.

Response Figure R10: Simulation domain and boundary conditions for the nanobeam cavity. The schematic illustrates the numerical simulation domain and boundary conditions applied to find the quasi-normal modes (QNMs). The blue shaded region is the silicon nanobeam, and the grey regions are the bowtie air holes and the surrounding air-box.

The simulation domain is meshed with different resolutions or mesh-element sizes in different regions, e.g., the silicon nanobeam region, nanobeam air holes, air-box surrounding, etc. Boundaries and edges around the bowtie region are meshed prior to the the rest of the geometry and with a higher resolution and a smaller value of the element growth rate than the rest of the structure in order to avoid discontinuities at the polygon's vertices. The precise COMSOL settings for the mesh in the regions specified in Fig. R11 are summarized in Table R1, all of them parameterized by a variable m which we vary to study the convergence of the fundamental cavity mode parameters. The resulting mesh for $m = 5$, the finest mesh we explore, is illustrated in Fig. R11. Figure R12 shows the mesh convergence results for the nanobeam cavity with the design parameters, i.e., that from Fig. 2 in the main text. The mode volume, resonance wavelength, and quality factor are plotted for varying m ,

Geometric Entity	Maximum element size [nm]	Minimum element size [nm]	Maximum element growth rate
Silicon nanobeam	$\lambda_r/n_{\text{Si}}/m$	$\lambda_r/n_{\text{Si}}/5/m$	1.6
Boundary 1	$5/m$	$0.5/m$	1.4
Boundary 2	$10/m$	$1/m$	1.4
Edge	$10/m$	$1/m$	1.4
Air box surrounding	$\lambda_r/n_{\text{Air}}/m$	$\lambda_r/n_{\text{Air}}/5/m$	1.7

TABLE R1: Mesh element sizes for different geometric entities of the nanocavity geometry. The sizes in the different regions are parametrized with variable m , which we vary to study numerical convergence. The value used for the reference wavelength, λ_r , and the refractive indices of silicon and air, n_{Si} and n_{Air} , are 1520 nm, 3.48 and 1, respectively.

with the axis of the latter on a log scale. The insets show the plots for m values greater or equal to 1, with the dashed lines illustrating the upper and lower bounds of the standard deviation of the values in the highlighted region. For our model, the mesh converges around $m = 1$, and we consider the deviations seen above that to be numerical noise, as most of the values fall within the bounds defined by the standard deviation. Our cavities in the manuscript are all simulated with the mesh parameter m within the converged range, typically at $m = 1$ for which a simulation of a single QNM (the one of interest) takes approximately 1.5 hours to run and consumes 150 GB of memory. Longer times and larger memory consumption are required for the simulations including disorder since more than an eighth of the nanobeam is simulated.

In addition to testing the convergence of the cavity mode parameters relative to the mesh, we also performed a convergence test on the air box surrounding the nanobeam. This is critical to converge the quality factor Q as too small surrounding air boxes may lead to still large evanescent field tails at the boundary that is supposed to mimic the radiation condition. Figure R13 shows the convergence test for the air box performed with $m = 2$, where the parameter L defines the height and depth of the air box as shown in Fig. R10. The cavity parameters converge for a surrounding air box of $L = 1.4 \mu\text{m}$, as most values are within the bounds defined by the standard deviation (same criteria as for the mesh size). We have used $L = 2.5 \mu\text{m}$ throughout the manuscript to model our cavities, which is well within the converged range.

- The resonant wavelength increases on decreasing the air gap within the bowtie tips, as clearly shown in Fig. 4d. I am sure this is correct, as it is also reproduced from COMSOL simulations, however, can the authors give an argument for why this is the case? In fact, intuitively one might think that reducing the air gap, and

Response Figure R11: Mesh elements for bowtie nanobeam cavity. a, Mesh elements for one-eighth of the nanocavity geometry with the converged value of the mesh parameter, $m = 5$. b, Mesh elements for different nanobeam cavity geometry domains, boundaries, and edges. The respective meshed regions are represented by the blue color.

decreasing the mode volume, corresponds to increasing the field intensity in the void (i.e., the lower refractive index) region, thus reducing the effective mode index, which should actually lead to a blue-shift of the mode... I am sure this argument does not apply, but I cannot figure out where I'm wrong here. Just a comment might be useful.

We appreciate the comment by the reviewer and asked ourselves the same question in the course of our research leading to the present manuscript. There are two effects in play here:

Response Figure R12: Mesh convergence test for a nanobeam bowtie cavity. Convergence test and converged values for the mode volume, resonance wavelength, and quality factor using different values of the mesh parameter, m , plotted on a log scale. The insets illustrate the log-scale plot of the regions highlighted with a black rectangular outline. The dashed lines represent the upper and lower bound of the standard deviation for the values in the highlighted region.

1. A reduction of the air region in the regions with high field intensity, which can be simply understood to lead to a red-shift as for first-order perturbation theory, i.e., $\Delta\omega \propto -\frac{\omega}{2} \int_V \Delta\varepsilon(\mathbf{r}) \mathbf{E}(\mathbf{r})^2 dV$, given that $\Delta\varepsilon(\mathbf{r})$ is always positive when the gap, g , is reduced.

Response Figure R13: Convergence test for the dimensions of the surrounding air box. Convergence test for the air box surrounding and converged values for mode volume, resonance wavelength, and quality factor plotted on a log scale. The air-box height and depth represented by the parameter L are varied to perform the convergence test. The convergence test was performed with the mesh parameter m of 2. Insets illustrate the log-scale plot of the regions highlighted with a black rectangular outline. The dashed lines represent the upper and lower bound of the standard deviation for the values in the highlighted region.

2. An extremely pronounced field-enhancement effect in the air bowtie as g is reduced, which might lead to a breakdown of the first-order perturbation theory approximation. If it were to fail, then a blue shift could be expected.

The reality is that the first-order approximation holds well (we have verified it) and what actually determines the effective refractive index of the mode is not the maximum/central field in the air bowtie, which determines the mode volume V , but the integrated energy density in it (or overall in

Response Figure R14: Confinement characteristics of the bowtie photonic-crystal nanobeam. Confinement factor, Γ , as defined in Eq. R5, as a function of the bowtie width, g , for the Bloch mode of interest at $k = \pi/a$.

the full air region) compared to the total energy density, i.e., the confinement factor,

$$\Gamma = \frac{\int_{\text{air}} \varepsilon(\mathbf{r}) \mathbf{E}(\mathbf{r})^2}{\int_{\text{total}} \varepsilon(\mathbf{r}) \mathbf{E}(\mathbf{r})^2}. \quad (\text{R5})$$

The value of Γ decreases with decreasing g for the cavities we present in our work, as evidenced for the Bloch mode at $k=\pi/a$ (Fig. R14).

- How can the spring constant of each structure be actually measured? The results are reported as a function of k/A , which is the relevant parameter, e.g., in Eq. S4; but which Area are the authors assuming to normalize the elastic constant? Or are they actually determining k/A directly? And in which way?

The two terms entering the expression k/A are the spring constant k of the guided folded cantilever spring system and the area A of the two parallel surfaces where the surface forces dominate, i.e., for platforms of width $w = 8 \mu\text{m}$, A is equal to $8 \mu\text{m} \times 0.22 \mu\text{m}$, with $0.22 \mu\text{m}$ being the nominal device-layer thickness, confirmed by tilted SEM images. The joint variable k/A is used based on the assumption that the surface forces acting upon the suspended plates can be understood as a pressure, which obviously neglects edge effects or any size-dependent fabrication effects but is consistent with the fact that we find platforms of different widths to lead to similar thresholds for self-assembly.

The values of the spring constants are extracted by simulating their force-displacement curves in COMSOL Multiphysics using the full anisotropic elasticity matrix for silicon [13], the correct crystalline axis orientation of the silicon-on-insulator wafer, and the SEM-extracted geometries of the fabricated springs. Figure. R15a shows the geometry of the device used to extract the spring constant values for a folded-guided cantilever spring system with a cantilever length, L , of $6.4 \mu\text{m}$. The geometry consists of a silicon platform attached to two folded guided cantilevers, thus, replicating our devices used in

Fig. 1 of the main text. A load, F , is applied on the platform in the direction of the y-axis as shown by the arrow in Fig. R15a, and the springs are anchored using the fixed-constraint boundary condition (BC). Figure. R15b shows the simulated displacement magnitude of the device at an applied load F of 96 nN. The force-displacement curve is plotted in Fig. R15c and a spring constant value of 0.56 N/m is extracted via a linear fit for a cantilever length L of 6.4 μm . We note that our simulations include the geometric non-linearity and therefore we expect the springs to be linear within a very wide displacement range. The spring constant values used in the experimental work are therefore extracted by repeating this process for varying cantilever lengths L .

In summary, the area A is extracted from the fabricated geometries and the spring constant is not measured, but simulated using the fabricated geometries. However, we have done extensive experimental and theoretical work on determining the spring constant of nanoscale silicon springs, which can be measured by applying an AC voltage to nanoelectromechanical actuators and measuring the resonance frequency. We have compared such experiments to several analytical models and find that none of them provides as good agreement with the experiment as the COMSOL model, which is why we use it here. We have reported on these details in a recent work [14] where we compare different types of nanoelectromechanical comb drives to theory and find that the differences between the measured spring constants and our COMSOL model are at most 34%. This turns out to have a very small effect on the plots reported here (Fig 1d in the main text, Supplementary Fig. S8 and Supplementary Fig. S9) because of the strongly nonlinear character of the surface forces and because, in any case, we base our design rules for self-assembly on the measured collapse thresholds, not the model.

Response Figure R15: Numerical simulations for the extraction of spring constants, k . **a**, Simulated geometry for the extraction of spring constants k . A load F is applied at the silicon platform in the y -direction, and the springs are anchored using the fixed-constraint boundary condition (BC). The folded guided cantilever length is defined by the parameter L . **b**, The simulated displacement magnitude, $|\mathbf{u}|$, of the device with an applied load F of 96 nN. Displacement is exaggerated for illustration purposes. **c**, Force-displacement plot gives a spring constant of 0.56 N/m for the simulated device with the cantilever length of 6.4 μm as shown in (a).

REVIEWER #2

This manuscript proposes and demonstrates a technique to form nm-wide, high

aspect ratio gaps and voids at the intended locations, which cannot be formed by direct patterning. This technique utilizes the pulling force generated at the gap to close the gap after the structure is released from the underneath SiO₂ sacrificial layer, where proper patterns are formed in the gap region in order to leave a small gap or void. Using this technique, they have formed an optical cavity with a bowtie structure that has a gap width of 1~3 nm and a gap height of 220 nm, and achieved a measured Q value of $1.5 \cdot 10^4$ and an estimated mode volume of $8.8 \cdot 10^{-4}$. This is highly evaluated as a new method to fabricate nm-level gaps and voids with good reproducibility.

We thank the Reviewer for considering our manuscript and for pointing out a specific aspect that is key in it: the self-assembly technique allows the formation of as many voids as desired at exact intended locations. This is a key differential aspect relative to other self-assembly approaches, where either periodic arrays are generated (so-called self-organization) or where the self-assembled entities are formed at random locations.

On the other hand, self-assembly methods using appropriate patterning and force after release have been conventionally utilized in the field of MEMS/NEMS. (For example; *J. of Microelectromechanical Systems* 12, 387-417 (2003), *small* 5, 1600-1630 (2009), and *Nano Lett.* 21, 3205-3210, (2021).) Although the scale and usage may be different from the one in this report, the basic concept of the method is not largely different from those in the field of MEMS/NEMS. Therefore, I think that the novelty of the technology in this report compared to those of the conventional MEMS/NEMS should be discussed to clarify the difference.

We have carefully studied the references suggested by the Reviewer and determined that they consider effects and mechanisms that are entirely different from and incomparable to our work.

We agree that self-assembly of pre-patterned compliant semiconductor structures by forces acting upon release or through the presence and processing (like melting) of additional materials is conventional in MEMS, with applications in optical, electrical, and mechanical MEMS, as thoroughly described in, e.g., Refs [15–17] – one of the references provided by the Reviewer. We realize that the previous version of the manuscript did not discuss the MEMS literature in detail. Space limitations as well as the request from Reviewer 1 to shorten our manuscript and also a request from the Editor to keep the number of references within the guidelines of Nature prevent us from elaborating in detail on this in the main text, but we now mention these aspects in the introduction of our manuscript.

A more detailed discussion follows below, which documents that regardless of the importance of self-assembly in MEMS, our method is entirely novel and the structures and devices we self-assemble could not be made with any of the previously known techniques.

The main self-assembly methods in MEMS are the following:

- **Capillary forces** have been used for self-assembly by locally depositing either solder bumps that are heated to re-flow [18, 19], or liquid droplets controlled by deposition of hydrophilic surfaces [15]. These techniques have been used in MEMS in conjunction with micromachined hinges to form three-dimensional origami-like structures; however, due to the inherent size of droplets, this technique has to our knowledge only been applied to structures in the micro- to millimeter size range. These methods of self-assembly differ fundamentally from ours, e.g., they require several additional process steps, such as multiple lithography steps and the deposition of metals, which incur losses in photonics, or the application of droplets that would collapse and destroy our suspended devices. Additionally, the application of droplets is generally controlled by a deposition step, making it non-trivial to adapt this technique for in-plane self-assembly and practically impossible to reach the subnanometer precision required for integrating atomic-scale structures with surrounding circuits. In contrast, our method offers inherent self-alignment. A slightly more stochastic approach has been used to self-assemble photonic structures, including individual plasmonic bowtie resonators[20]. Here the entire sample is submerged in ethanol, which is then dried to induce collapses. This approach is not limited by droplet size, and circumvents the need to control and guide the position of droplets. However, it can only be applied to substrates on which all devices are meant to collapse, making it inapplicable to our devices.
- **Unbalanced film stress** is known in MEMS to cause suspended structures to curl, and it has been used as a self-assembly method for bending structures out-of-plane [21, 22]. This method relies on the deposition of a material with a different thermal expansion coefficient than the bulk device material and bends devices out-of-plane. Such methods are therefore fundamentally different from and unsuitable for the in-plane self-assembly demonstrated in our work.
- **(Di)electrophoresis** uses electric fields to guide particles suspended in liquids. This technique is most commonly applied in biology to sort cells but has also found applications in micro- and nanotechnology [23]. While it is an intriguing fabrication method, its reliance on a liquid suspension is incompatible with our suspended structures (for which vapor-phase etching is crucial as discussed in our manuscript). Also, the interfacing with external circuits demonstrated in our work would be highly non-trivial or even impossible to do with electrophoretic self-assembly.

- **Casimir and van der Waals attraction** have been used to form organized clusters of particles in liquid suspensions [24], and have even been employed to form photonic cavities [25]. To the best of our knowledge, all previously reported uses of these attractive forces for self-assembly have either been mediated by liquid or resulted in devices where the liquid is crucial even after the self-assembly. Our method replaces the liquid with floppy springs, giving us complete control over the final position and orientation of our self-assembled devices while avoiding the need to apply liquids to our suspended structures. In any case, the self-organization of clusters cannot be used to self-assemble complex circuits.
- **Powered actuators** such as electrostatic [26], thermal [27], or magnetic actuators [22] could all be used to assemble in-plane structures, but typically require more extensive fabrication, and complex circuits would require an even more complex actuation circuit. In any case, these methods do not rely on self-assembly, which is the central aspect of our work.

In summary, the existing self-assembly methods in MEMS are fundamentally different from our method. Notably, both the achievable feature size and the absolute position accuracy of those features are orders of magnitude worse than what we demonstrate in our work. While these other methods offer many exciting applications for, e.g., self-assembled clusters or out-of-plane origami devices, they could not have been used for fabricating our devices. Finally, to the best of our knowledge, our work is the first to report on a method for in-plane self-assembly compatible with and self-aligned to surrounding complex circuitry.

Also, I think the claim in the title and introduction is a bit too large, since the technology in this report is basically to create gaps and voids at the nm level, i.e. a claim like “self-assemble of atomic-scale gap” may be appropriate.

We agree that the demonstration of atomic-scale gaps is central to our work, and the title was chosen in accordance with standards in the literature of photonic cavities, which are typically named after their defining geometric feature(s). For example, a photonic-crystal cavity is notably not photonic crystal but a defect in a photonic crystal, and a bowtie cavity is notably not a bowtie because a bowtie in itself is a broadband field-concentrator device turned into a cavity by the surrounding Bragg mirrors. We would also like to mention that while positive features approaching the atomic scale could in principle be made by digital etching of larger structures, it is generally much harder to make small gaps, and in fact the atomic-scale gaps demonstrated in our work could not be realized by any other known fabrication method. We also want to stress that our concepts have a much broader significance than just making small gaps: We foresee that our

methodology will also enable the creation of nanometer-scale features in materials other than air. These can be deposited by, e.g., atomic layer deposition (ALD) before the self-assembly process. In such a case, the pre-underetch distance between the self-assembled parts will be large enough to enable conformal growth of the ALD layer, which can form the sub-nanometer feature upon collapse. As an alternative, materials can be deposited after self-assembly using ALD to form a nanometer-scale bridge for which the dimensions will depend on the gap between the bowtie tips and the deposited thickness. SENTENCE DESCRIBING UNPUBLISHED MATERIAL REMOVED. Taking the Reviewer's comment as well as the considerations above into account, the central aspect of our work is the atomic-scale confinement, not the size of the cavity, and we have modified the title in the revised submission to "Self-assembled photonic cavities with atomic-scale confinement". We actually find this title more precise and thank the Reviewer for directing our attention to this point.

Response Figure R16: CONTENT CONTAINING UNPUBLISHED MATERIAL REMOVED

However, the fact that a gap of about 1 nm was formed with high fabrication accuracy is highly commendable. In this regard, it is thought that an extremely vertical etching profile is required to apply this method. Therefore, I think that the relationship between verticality and the gap formed should be discussed. In addition, it is necessary to discuss whether a special method is necessary for achieving such high verticality, as well as the reproducibility and pattern dependence of the verticality.

We agree that the fabrication of extreme-aspect-ratio nanometer gaps is related to the sidewall verticality. Upon reading our manuscript with the Reviewer's comment in mind, we realized that the discussion of these aspects was too brief in the previous version of the manuscript, where we mostly discussed them in the context of the extracted sidewall angle in the STEM image analysis in the main text.

The exact way in which the two etched surfaces of the spring-suspended parts end up after a stable equilibrium is reached depends on the verticality of the produced sidewalls before underetching. Ideally, the sidewalls are perfectly vertical and parallel, such that the net force has no out-of-plane component (even in the presence of fringing fields [28]). In this case, the surfaces, when they collapse, would be perfectly aligned. However, sidewalls rarely have a perfect 90° angle. Since the strength and direction of the surface forces, e.g., Casimir-van der Waals forces, depend on the exact geometry of the gap between the two parts [29], the force landscape depends on such sidewall verticality and therefore determines the pull-in instability threshold. As we mention in our discussion of the STEM images, we observe a relative angle between the crystalline silicon lattices on both sides of the nanobeams between 1° and 2° , and generally find no visible trace of gapped areas across the collapsed regions - at least within the resolution of the SEM images. In addition, such an angle agrees reasonably well with the double of the complementary angle of the sidewalls that we observe on 50, 90, 120 and 200 nm wide trenches in an etch test chip (see Fig. R17) fabricated with the same process but two weeks before the fabrication of the sample containing the cavities reported in Figs. 2 and 3 in the main text of the manuscript. We note that the small difference in crystal orientation of the two halves of the cavities observed in the STEM images indicates that such minor sidewall inclinations have a very minor effect on the resulting devices because the compliant halves simply bend by the same angle out of plane, such as to keep the interface parallel. This would in turn result in a slightly higher transmission loss and reduced quality factors, which we indeed observe, cf., the response to Reviewer 1 and the revised SI Section 5.2.

In the initial submission, we had decided not to include much detail on the nanofabrication process and instead refer the readers to Refs [4, 30], which describes it extensively. Nevertheless, considering that the verticality of the sidewalls has been mentioned by both Reviewers 2 and 3 and that the self-assembly process we propose allows, first, avoiding the lag associated with the reactive ion etching as well as the loading effects that occur in narrow gaps [31, 32] and, second, the realization of self-assembled and self-aligned features, we have decided to include the detailed information in the SI Section 4.1 in the revised SI that shows the cross-section SEM images reproduced here as Fig. R17.

There are also several questions regarding the optical cavity part. First, in Nat Commun. 13, 6281 (2022), the same group reported that a mode volume of $3 \cdot 10^{-4}$

Response Figure R17: Etch profile of trenches in silicon using chromium as a hard mask. a, Etching profile of 50 nm wide trenches. b, Etching profile of 90 nm wide trenches. c, Etching profile of 120 nm wide trenches. d, Etching profile of 200 nm wide trenches.

and a measured Q value of 1000 can be achieved without nanogaps and with a very complicated structure. We would like to know the relationship between that structure

and the structure in this manuscript.

Our previous work on bowtie nanocavities differs considerably from the one presented here in several regards. The cavity in Ref. [11] was designed using topology optimization and targeted the local density of optical states at a single point, i.e., a single-hotspot cavity. In addition, the in-plane footprint was limited to $4\lambda^2$ due to the computational load required for density-based inverse design and to develop a compact standalone cavity. This, in turn, limited its Q -factor to values around 10^3 or below. We note that what the Reviewer refers to as "a very complicated structure" does not contribute much to the value of the mode volume, which is set by the bowtie width, but rather to the cavity Q , i.e., the complicated structure acts as Bragg mirrors. On the contrary, the cavities we present here take a different approach based on photonic-crystal nanobeam cavities with bowtie unit cells. This allows targeting much larger quality factors (at least one order of magnitude more) by using tapered defects. More importantly, the fabrication constraints, such as the minimum feature size achievable with our top-down nanofabrication, were considered during the cavity optimisation process in Ref. [11] and led to an 8 nm silicon bowtie width. Such a value makes the etching very challenging (the etching easily erodes the 8 nm bridge), and we had to sacrifice sidewall quality to improve etching selectivity, which resulted in the so-called scallops (periodic protrusions on the sidewalls) visible in the SEM images in Ref. [11]. In the present work, we have no such limitations because we fabricate our devices safely within the borders of our fabrication constraints and let the self-assembly make the small features. The net result is that our self-assembly method allows at the same time fabricating much smaller features than in Ref. [11] while at the same time being much easier to fabricate. In addition, we have used a more advanced fabrication process using a chromium hard-mask, which provides excellent selectivity and gives further improvements to the sidewalls, cf. our response to the previous question.

By the way, in Fig. 5, the δ is 12 nm, but in accordance with the description of L347-348, it seems that the gap is closed in this case

We are sorry if this point was not made clear in the previous manuscript version. The waveguide-coupled self-assembled cavities in Fig. 4 in the revised main text (previously Fig. 5) and the standalone cavities in Figs. 2 and 3 (previously Figs. 2, 3, and 4) have different overall layouts (because they are either standalone or in-line coupled) and belong to two different samples fabricated 7 months apart. These two aspects are important in determining the exact offset-to-width correspondence, $g(\delta)$. As we discuss in Section S4.2 in the SI, the bowtie width after self-assembly depends on the offset, δ , but also on a number of process-dependent fabrication parameters. In the SI, we mention two factors: the

critical-dimension loss, Δe , and the solid radius of curvature, R_S , and try to model the relation $g(\delta)$ based on these two (see Eq. (S6) in the SI). We observe and intuitively understand that the first of these two parameters can vary from sample to sample. There are two reasons for this variation:

- Proximity effects:** Even if we are using advanced proximity effect correction (PEC) on the lithographic mask using Beamfox Proximity, we observe that the size of specific features, e.g., the gap and triangle in the nanobeam, depends on the size of other neighbouring exposed areas, e.g., the trenches that define the nanobeam. The reason is that no proximity-effect correction algorithm has so far been able to rigorously capture and correct for the short-range proximity effects, and while these effects are irrelevant in many cases, they are certainly important when fabricating devices whose ultimate (self-assembled) critical dimension drops below one nanometer. This could be alleviated by improved proximity-effect correction algorithms that would take the interaction of electrons with the full silicon-on-insulator stack as well as process correction fully into account, i.e., including the etching of the hard-mask consisting of chromium, poly-silicon, and resist and the final silicon etching and other processes. With this in mind, an inspection of the lithography mask reveals the origin of the differences between the different devices: The cavities for resonant scattering measurements consist only of a nanobeam cavity and folded guided cantilevers. On the other hand, the waveguide-coupled cavities have a much larger exposed region around them, consisting of a tapered section, circuit crossings and long waveguide segments, including grating couplers. The extra dose which comes from exposing these components makes Δe larger for the second sample even if all the rest of the parameters were fixed, and therefore shifts the straight line $g(\delta)$ up.
- Fabrication-process drift and fluctuations:** We observe Δe to drift and fluctuate over time as the process itself drifts. This is unsurprising given that we fabricate our devices in a shared university cleanroom with lithography and etching tools that run a large variety of processes. Generally, we observe Δe to grow when either the lithography or dry-etching parameters depart from optimal conditions, which occurred when fabricating the chip used for the waveguide-coupled cavities. An indication of that air-region growth is given by the resonant wavelengths of the waveguide-coupled cavities, which are considerably blue-shifted relative to the standalone resonant-scattering ones.

To summarize, on the chip we use for the waveguide-coupled cavities, the measured critical dimension loss Δe , evaluated through the change in the width of the gap from the lithography mask to the pre-underetched sample, was ~ 13 nm, which is considerably more than on the chip we use for standalone cavities (~ 3 nm). If these values are inserted into Eq. (S6), we find that the offsets, δ , for which the gap closes are 9.5 nm and 5.4 nm, respectively. Our simplified model fails to accurately

predict the offset at which the gap closes, but it does show that we expect the waveguide-coupled cavities to exhibit the same gap as the standalone ones for larger values of δ . The importance of the final gap width on the batch-specific parameters is the reason why we decided that the offset, δ , is spanned over a wide range of values in the chips we have explored. Nevertheless, one could expect that a more process-specific cleanroom environment and advanced proximity effect correction could allow any arbitrary bowtie width down to zero. We elaborate more on these points in our response to Reviewer 3.

Also, in Fig. 4f, the mode volume is the smallest when the gap is closed. If so, to what extent does this technology, which can reproducibly create a nanogap, contribute to the realization of a very small mode volume?

The point raised by the reviewer is quite interesting since it would appear as if the mode volume is smallest when the gap is closed. Nevertheless, we note that Fig. 3b of the revised manuscript (previously Figs. 4 d, e and f) did not go down to an air bowtie width of 0 nm but only to 0.25 nm. The reasons for not including 0 nm are various:

1. Getting the gap down to 0 nm implies that the geometry has a singularity where the two rounded tips are tangential to each other. The geometry is, therefore, far from realistic, since the true geometry would have to consider roughness and probably some atomic rearrangement.
2. The presence of such a sharp (double) tip leads to lightning-rod effects, i.e., divergences in the electromagnetic field, as we have extensively discussed in previous work [12]. We evaluate the mode volume at the center of the central bowtie (as described in the new Section S3.2 of the Supplementary Information), which for a gap of 0 nm coincides with the boundaries and the diverging point. This might lead to a numerical artifact in the evaluated V .
3. The numerical solver employed (COMSOL Multiphysics) fails to even find the relevant cavity mode for the case of 0 nm bowtie width when the oxide is not present. We do not understand why that happens and why, instead, the solver finds many numerically spurious modes around the expected wavelength, but for the reasons mentioned above, the limited case of 0 nm gaps is not physical.

For clarity, we have tried to *regularize* the geometry for the cases where $g = 0$ nm, both with and without the oxide, which transforms the geometry as seen respectively in Fig. R18a and Fig. R18b. Figure R18c shows the wavelength, quality factor, and mode volume calculated for the cavities with no oxide and 2 nm oxide width as the bowtie width, g , is varied. The plots include the regularized geometries represented by the horizontal dashed lines. We note that, to our surprise, when the oxide is present, the figures of merit of the un-regularized $g = 0$ nm fundamental cavity mode (red open

circles) line up quite well with the rest of the data, even if their values have much stronger mesh dependence. On the contrary, the mode is not found in the absence of the oxide, as stated above. Interestingly, for the more realistic regularized geometries, we see that the figure of merit discussed by the reviewer, V , slightly goes up for the case with oxide and dramatically goes up in its absence. Therefore, a more realistic setting for the collapsed geometry with $g = 0$ nm leads to larger V than finite but very small air gaps.

Response Figure R18: Quasi-normal-mode characteristics for vanishing gaps in regularized geometries. a, Superimposed geometries for a bowtie width $g = 0$ nm before and after regularization in the absence of the oxide layer. The added material is silicon. b Same as a for a geometry with an oxide layer of width $d = 2$ nm. The added material is silicon oxide. c Variation of the wavelength, quality factor and mode volume as a function of g without (top) and with (bottom) the oxide. The horizontal dashed lines represent the values for the regularized geometries at $g = 0$ nm. The values for singular geometries at $g = 0$ nm for the oxide case are highlighted in red.

Regardless of these numerical technicalities: if one were to believe the numbers found when $g = 0$ nm with the two perfect rounded tips colliding, they would still be impossible to achieve without the self-assembly approach. Conventional top-down fabrication would be limited by the finite solid

and void radii of curvature and would not allow, by construction, such a geometry. In addition, when the oxide is present (the only case for which we have the guarantee of the optical mode of interest existing as described above), light confinement occurs due to the presence of a 4 to 6 nm wide low-refractive index medium, silicon oxide, which would also be impossible to realize with other conventional techniques. In conclusion, the small gaps do not only contribute to the small mode volume, they are absolutely instrumental to reach the confinement levels discussed here and presented in our manuscript. To the best of our knowledge, the only method enabling the construction of devices at this scale is the self-assembly protocol presented for the first time in the present manuscript.

Perhaps it is easier to create a gap of 0-1 nm using the method in this report than to create a continuous bridge of 2 nm width in the previous report.

Yes, it is in fact much easier. Generally, it is more difficult to create void features with a high aspect ratio than high aspect-ratio solid bridges or pillars using lithography and etching only, but this is exactly what our methodology circumvents, because our devices are fabricated with dimensions well inside the bounds of our fabrication constraints for lithography and etching, and only the self-assembly step goes below the fabrication constraints. In contrast, a 2 nm solid bridge is way below our fabrication constraints, which we have measured with great care and reported in our previous report Ref. [11], and may only be achieved at the price of a very poor sidewall quality (which has been the main problem in previous attempts than Ref. [11] at making solid dielectric bowties). Already the 8 nm silicon bridge in our previous work required extensive manual shape corrections of the lithography mask: We refer the Reviewer to pp. 4-9 in the SI of Ref. [11] where this tedious procedure is explained in detail. Although digital etching could in principle be employed to thin down 8 nm bridges to 2 nm, this would come at the expense of reduced resolution, unwanted spectral tuning, and it is not even obvious how this would be achieved in practice because digital etching removes 2 nm of surface oxide, so digital etching may likely be too crude a method to dimensions of 2 nm, let alone the much smaller dimensions demonstrated in our work using self-assembly.

Also, I think the method and structure in this report is better in terms of the possibility of introducing a light emitter, etc. However, I would like to see a discussion of this issue.

Yes, we agree with the Reviewer, and the main text briefly discusses the following. We would like to point the Reviewer's attention to the current version of the conclusion for the discussion on introducing light-emitters and exotic nonlinear materials at the bowtie location. One of the main

drawbacks of silicon photonics is the absence of efficient emitters due to the indirect bandgap, and our method opens perspectives for integrating emitters operating in the telecommunication C-band, such as erbium ions. Actually, the luminescence of erbium in silicon oxide is much stronger (and much more studied as it is the workhorse material system for optical amplifiers), so our glass-core cavities where the bowtie tips are touching to form a glass interface between the bowtie tips due to the native oxide layer and having erbium emitters could offer an unprecedented enhancement of the photoluminescence with a Purcell factor in the range of millions. Using atomic layer deposition, erbium or similar emitters can also be deposited between the bowtie tips before or after the self-assembly [33]. Such cavities could also turn out to be essential for applications such as single-photon or multiple-photon non-linearities in, e.g., silicon, which has a rather weak Kerr non-linearity and non-existent χ^2 non-linearities. The paper by Choi et al. [34] suggests that embedding nonlinear materials, such as ITO, TiO_2 , Si_3N_7 with χ^3 or polymers with very large χ^2 non-linearities, between the bowtie tips can massively enhance the nonlinear response. More generally, since our methodology is material agnostic, any combination of host material and embedded material is possible, limited only by the availability of standard etching and deposition methods. We, therefore, envision our work as the first steps towards integrating novel materials such as silicon nitride, silicon carbide, or lithium niobate with high-aspect-ratio single-nanometer semiconductor structures made from silicon or other relevant host material, and enhancing the exotic properties by using the extreme confinement offered by our method.

Another excellent demonstration in this manuscript is to make the resonator part and the coupling waveguide part separately and attach them together by self-assembly.

We thank the Reviewer for pointing out the important aspect of integrating self-assembled cavities with photonic circuits, which is critical for applications where efficient in- and out-coupling of light is required. The integration we demonstrate is also essential because it shows, for the first time, how the one-time self-assembly step, typically considered to be disjoint from the interconnect architecture, can be carried out at the same time as the rest of the (in this case photonic) circuitry, which is actually crucial, because only self-alignment is able to align the circuitry with the atomic-scale features in our devices.

Considering all the above, I think that the value of the results in this manuscript is in the fact that it was actually possible to create a very fine structure with good reproducibility, and the fact that a bowtie cavity with a small gap and good optical characteristics was fabricated using this method. Although it is difficult to judge, I am afraid that this manuscript at present form is more suitable for more specialized journals

such as Nature Nanotechnology and Nature Photonics than Nature.

We are convinced that our manuscript meets the publication criteria for Articles in Nature. To briefly summarize the novelty and impact of our work:

- We demonstrate a novel self-assembly method, which is unlike any techniques in the literature. Our method is novel, robust (fabrication yield $> 99\%$), and applicable to all material platforms where selective underetching is available.
- We show that our method can be used to realize devices with unprecedented dimensions and aspect ratios: Our devices are impossible to make with any other method and their critical dimension is an order of magnitude below the most optimistic projection at the end of the semiconductor industry's lithography roadmap.
- Our method is capable of making complex devices that are not only self-assembled, but also self-aligned to the surrounding circuitry and thereby bridging the atomic and macroscopic scales, which was never done before and is a very significant advancement in self-assembly.
- Our mapping of the conditions for self-assembly rest on extended data sets measured on thousands of devices, which is well beyond the statistics underlying the vast majority of papers in nanophysics, nanotechnology, photonics, nanoelectromechanical systems, and related fields.
- We illustrate the versatility of our method by fabricating the smallest optical cavity ever realized in a semiconductor system.
- Our work crosses boundaries between and connects photonics, Casimir-van der Waals physics, nanotechnology, semiconductor devices, nanomechanics, and nanocavities. While some of these fields are commonly connected in the literature, we are not aware of any works that bridge all these fields, and we are convinced that this will result in an exceptionally wide impact.

In summary, we report original scientific research with conclusions of outstanding scientific importance for an interdisciplinary readership, which are the publication criteria for Articles in Nature.

REVIEWER #3

Babar et al. describe their work on “self-assembly of atomic-scale photonic cavities” using van der Waals attraction and pull-in instability. The manuscript is well-written and presents an intriguing concept for the nanofabrication of photonic devices made of underetched silicon. The authors even present the far-field cross-polarized optical characteristics of these cavities and their integration with photonic circuitry. However, I have some concerns about the technical points presented, which prevent me from recommending this work for publication in Nature at this time.

We thank the Reviewer for the overall positive description of our work and of the manuscript. We hope that our response below resolves the Reviewer’s concerns about the technical points.

1) Firstly, the most critical point that needs to be addressed is the resolution of the final gap in the bowtie regime. While it is clear that surface roughness can determine the gap for the case of van der Waals contact of flat surfaces (as demonstrated in Fig. 1), for the case of the bowtie nanocavity, the gap should not be determined by surface roughness (r) as the authors suggest, but by the offset (δ) between the flat region and the bowtie region during the collapsing of the two nanofabricated halves. This offset, in turn, is determined by conventional nanofabricating tools such as electron beam lithography and reactive-ion etching used in this study, which could lead to substantial uncertainty in the fabrication of large-scale devices, impacting scalability and reproducibility. To explain this further, imagine taking two plates that are both rough and nanopatterned and then collapsing them. In the approximation of hard bodies, the geometrical constraints do not allow for a higher precision gap region than the offset (δ). Although the native SiO₂ formed at the surface may vary in thickness and roughness, it will be the same in both the flat and bowtie regions. It is unclear how the resolution of the final gap region can exceed the standard resolution of nanolithography used to produce the halves. Therefore, the authors should provide convincing arguments or a cartoon to explain this, or else adapt their message accordingly. While the authors may have been able to “fish out” bowties with sub-nanometer gaps in thousands of fabricated devices, it is unrealistic to claim that this can be scaled up massively, as the introduction and abstract seem to claim (the uncertainty in the fabrication of large-scale

bowtie devices will be still determined by standard nanofabrication).

These comments address central parts of our work and, and, since these questions are all connected, we hope the Reviewer will forgive us for answering them in a slightly different order and begin with a more general explanation.

Starting with the question of how our method allows making devices with critical dimensions much below the resolution of lithography and etching (despite using only lithography and etching), it is important to clarify what is meant by resolution in nanofabrication. We use the term “resolution” following the definition used in the semiconductor industry, where it denotes the smallest critical dimension, i.e., the width of the smallest line that can be reliably fabricated, e.g., 12 nm for the current “3 nm” node with an ambition to reach 8 nm at the end of the roadmap in 2037 [35]. Strictly speaking, the numbers in the roadmap are half-pitches of very large arrays of equally sized positive (ridges) and negative (gaps) lines, i.e., the half-pitch is the same as the width of a single line. In our work, we refer to single gaps rather than arrays of gaps and lines but since it is much harder to fabricate gaps than ridges, this is a fair comparison. The reason why gaps are harder to realize is in part that it is much harder to do proximity-effect correction for small gaps than for small ridges because it is always possible to add more electron dose to a ridge to bring the exposure above the clearance dose. In contrast, it is not always possible to remove enough dose from a gap to bring it below clearance (if the surrounding features require so high dose to be cleared that the proximity dose in the gap is nearly at clearance). In addition, micro-loading effects during dry etching are obviously also much more severe for small gaps than for small ridges (because the ridges can be surrounded by large features for which the micro-loading effects can be made small). Importantly, we demonstrate explicitly STEM measurements that our method allows building devices with gaps that are more than an order of magnitude smaller than the resolution of state-of-the-art CMOS production today, although we only use technology that is available already today in most smaller foundries and well-equipped university cleanrooms. Obviously, our self-assembly protocol requires the devices to be underetched, but for nanophotonic and nanomechanics, this underetching is only an advantage (e.g., improved optical confinement in nanocavities and silicon photonics) or even a requirement (nanoelectromechanical systems, gallium-arsenide quantum photonics for which there is no low-index substrate, etc.). In summary, our self-assembly method certainly allows making features well below the resolution limit in both industry and research despite using only conventional fabrication techniques. This is directly evidenced by our devices showing gaps below 1 nm, which should be compared to the state of the art today in the “3 nm” node, namely 12 nm, or, the “0.5 nm eq” node at the end of the industry’s roadmap in 2037, namely 8 nm.

The remainder of the question touches upon the interplay between subtle aspects of nanofabrication,

Response Figure R19: Schematic illustration of the deviations between lithography mask and resulting devices and why the gaps in our self-assembled cavities independent of the resolution.

a, A gap consists of two parallel silicon edges. The lithography mask (solid lines) is transferred into the silicon with two kinds of unavoidable errors: Structural disorder of mean zero (random lines) varying around a global shift (dashed lines). The resolution (not shown) is defined as the smallest linewidth that can be reliably fabricated, and the combined effect of the global shift and the structural disorder determines the total error of devices made exactly with the smallest possible critical dimension. b, When fabricating bowtie cavities by our method in a foundry setting, the mask would be designed within the boundaries of the fabrication constraints, i.e., with roundings reflecting the smallest possible solid and void radii of curvature. This means that the global shift moves the silicon edge in a conformal manner while the structural disorder is a random function. c, The same structure after self-assembly results in a gap determined by the offset between the planar sides and the tip, but this cannot be made smaller than the disorder, which, for a given fabrication process, limits the smallest possible devices that our method allows fabricating. In today's foundries, both the global shift and the amplitude of the disorder are below the size of a single silicon atom.

in particular between mask dimensions, process variations, and surface roughness. We refer to Fig. R19 in the present document for a cartoon illustrating these points. Figure R19a illustrates the effect of these parameters when aiming to fabricate a single trench using lithography and etching before/without self-assembly. Note that we are here assuming that the mask is designed within the limits of the fabrication constraints such that it does not contain, e.g., sharp corners. We refer to the discussion of

the critical radii in the SI Section 4.2 for additional details on design rules and fabrication constraints. By obeying the design rules, the edge of the fabricated line in the fabricated device will be exactly at the mask edge except for two undesirable effects: a random local variation with zero mean due to surface roughness and a global shift due to global drifts in parameters for the particular fabrication batch (due to, e.g., day-to-day variations in resist thickness, developer temperature, gas flows in the dry etcher, etc.). Obtaining precise models for the disorder is very complex, and we refer to our response to Reviewer 2 for a detailed discussion, but for the present question, the exact details of the disorder are not important. Rather, the present question is how the local disorder and the global shift affect our self-assembly protocol, how large these two effects are, and how our method may be scaled up. In this context, it is important to distinguish between what we demonstrate in our manuscript using fabrication in a shared university cleanroom and what could be achieved using dedicated foundry fabrication. As we document in the following, today’s foundry production allows reproducible placement of lithographic boundaries with near-atomic precision, which implies that reproducible and scalable production of devices based on our methodology is directly available in foundries today.

Figure R19b shows the effect of local disorder and global shifts when fabricating a bowtie structure, which is afterward self-assembled into the cavity shown in Fig. R19c. It is important to stress that the mask shown in Fig. R19b differs from the mask in our experiment on an important point: It obeys the design rules. In our experiment, our mask pattern employed sharp tips in an attempt to get the smallest possible solid radius of curvature in the final devices, but in a foundry, the shape would be designed within the borders of the design rules. This means that there are additional effects of mask erosion and proximity effects at the tips of the bowties in our particular realizations that are by no means fundamental to our concepts. When operating within the borders of the design rules, the global shift including the oxide will be conformal, which is also in line with the Reviewer’s comment: “Although the native SiO₂ formed at the surface may vary in thickness and roughness, it will be the same in both the flat and bowtie regions.” Regardless of whether the devices are made in a university cleanroom or produced in a foundry, the important point of our self-assembly protocol is that the final gap does not depend on the global shift of the silicon edges and neither does it depend on the resolution: The gap depends on the offset (but can be fine-tuned by other parameters as discussed below) and it is ultimately limited by roughness. This is the statement we make in the main text of our manuscript: “[...] the ability of our method to build atomic-scale semiconductor devices in which the critical dimension is limited by structural disorder rather than lithography.” We do not write that the gap is determined by the disorder (which we agree would be incorrect; indeed the offset and other factors determine the gap), but we write that the ultimate limit for devices made by our method is set only by the amount of structural disorder – entirely independent of the resolution. Having said

that, we acknowledge that the sentence quoted above could potentially lead to confusion because the structural disorder depends on details of the fabrication including the lithography. In the revised main text, we define explicitly what is meant by resolution, and we have replaced "lithography" with "resolution" in the sentence above, such that it now reads: "[...] the ability of our method to build atomic-scale semiconductor devices in which the critical dimension is limited by structural disorder rather than the resolution."

Regarding the role of the offset, δ , we agree that it is the main factor determining the gap in our self-assembled cavities. In the absence of deviations between the lithography mask and the final device, it would be the only determining parameter (as discussed in the SI Section S4.2). Although we only vary the offset in our experiments, it would be straightforward to also locally vary the electron-beam dose as well as a number of additional process parameters such as development time and temperature, etching parameters, etc., which would add continuous control over the resulting gaps. The gaps can therefore be made arbitrarily small – although not smaller than the surface roughness as discussed above.

While we do not fully agree with the connotations of the term "fishing out," because we present a highly systematic study with explicit discussions of the yield, we acknowledge that there is a finite variance of the gap size in our fabricated devices, which is shown implicitly or explicitly in several figures in the main text and the SI. However, we would like to stress that although the variance is significant on the atomic scale, it does not prevent us from realizing dimensions that surpass the resolution of state-of-the-art nanotechnology, as discussed above. We would also like to point out that the second part of the comment, "[...] the authors may have been able to "fish out" bowties with sub-nanometer gaps in thousands of fabricated devices, [...]" is not a fair description of our work. We do indeed characterize thousands of collapsed platforms to rigorously map the self-assembly phase space. Still, the number of cavities is much smaller: For example, we have done STEM measurements on 5 cavities with offsets $\delta = 8, 9, 10, 11,$ and 12 nm and already find the presence of sub-nanometer gaps or gaps that are just limited by roughness (see Supplementary Fig. S23), and this indicates a much higher yield than one to a thousand.

Now, let us turn to the question of how our results open a path toward a scalable process for making bowtie nanocavities with sub-nanometer gaps. Or, more specifically, why we stand by our statement in the abstract about our work constituting "the first steps towards a new generation of fabrication technology that combines the atomic dimensions enabled by self-assembly with the scalability of planar semiconductors". We have already explained that the limiting factor is surface roughness, so the question is what the roughness would be in a scalable production setting, i.e., in a semiconductor foundry. It is also clearly demonstrated in our work, explicitly in Fig. S19 of the revised SI, that we can systematically vary the gaps from positive to negative, so the remaining question is

Response Figure R20: Dependence of the resonant wavelength on hole radius. The resonant wavelength of L4/3 photonic-crystal cavities for 4 different hole radii (grey dots), which is equivalent to global shifts of the silicon edges. The error bars indicate the estimated numerical accuracy from a convergence test. A linear fit (solid line) gives a slope of -3.92 nm/nm.

with what precision the edge of the silicon can be defined. Here we refer to the recent work by Panuski et al. from the Englund group [36], which presents a study of photonic nanocavities fabricated in a foundry, notably using silicon membranes of the same geometry as ours. More specifically, Fig. 4d in Panuski et al. presents measured distributions of the resonant wavelengths and quality factors on large ensembles of several types of photonic-crystal silicon cavities from which the convolved effects of the global shift of the silicon edges, structural disorder, as well as all other deviations from the mathematically perfect device can be extracted. The measured standard deviations of the resonance wavelengths are found to be between 0.6 nm and 1.1 nm. It is easy to estimate how this translates into structural deformations by considering a simple example: If the silicon edge grows by 1 nm, the diameter of the holes in the photonic crystals grows by 2 nm, and this changes the optical path length by $\sim 2 \text{ nm} \cdot (n_{\text{Si}} - n_{\text{Air}}) \sim 5 \text{ nm}$, which shows that spectral measurements are extremely sensitive to structural shifts (which is of course not surprising given that the same effect underpins interferometric laser ranging, gravitational-wave detection, and much more). To be more precise, we consider the worst-case scenario of the L4/3 cavities studied by Panuski et al., for which the standard deviation in the resonance wavelength is measured to be 1.1 nm. The particular L4/3 cavity studied by Panuski et al. is a complex inverse design with a number of shifts applied to the hole position throughout the design in order to tailor the far-field, but the central part is a simple L4/3 design, which is simpler to simulate and it is reasonable to assume that the sensitivity to global shifts will be nearly identical across different variants of the L4/3 geometry. We have therefore simulated the L4/3 cavity discussed

in Ref. [37], and the result is shown in Fig. R20, which shows the resonance wavelength as a function of the radius of the holes which equals the global boundary shift. From a linear fit to the slope, we find that the shift in resonance wavelength per shift in silicon boundary is -3.92 nm/nm , which is consistent with but more accurate than our simple estimate above. Thus, the data by Panuski et al. imply that the silicon edge is defined within a standard deviation of $1.1\text{nm}/3.92 = 0.28\text{nm}$. This should be compared to the interatomic distance in silicon of 0.24nm , i.e., even if the resolution of the Panuski experiment is far worse than in ours because we use electron-beam lithography, the silicon edges in the Panuski work are defined within approximately \pm one atom, which is significantly better than what we can realistically do in our cleanroom. Even if this already shows that our claims are valid: Our method circumvents the resolution limitation, and the next limit is structural disorder, which already today is at the atomic scale in foundry production, we would like to add that we have made several unrealistically pessimistic assumptions here. In reality, other factors such as the global shift, variations in wafer thickness, chemical composition at the edges, etc. all contribute to the experimental value found by Panuski. et al., and this implies that the actual disorder is lower than our estimate. Also, we have considered the L4/3 cavity for which the standard deviation is the largest. In reality, both the definition of the silicon boundaries as well as the structural disorder in today's foundries are at or below atomic resolution, and this implies in turn that our method could be directly used to massively scale up the production of truly atomic-scale devices - even in a foundry where the resolution is limited to several tens of nanometers.

We hope that our explanations clarify the points raised by the Reviewer, and we hope that the Reviewer now agrees that the claims made in our abstract and in the main text are, in fact, very solid. Having said that, we realize that other readers may have similar questions, and we have made several amendments to the revised submission to address this. In our revised submission, we include the definition of δ in Fig. 2 in the main text, (the revised version of which can be found in this response as Fig. R1), which includes a very simplified version of the cartoon discussed above. Space constraints along with requests from both Reviewer 1 and the Editor to reduce the number of figures as well as the amount of text makes it hard to fit in more details on these points in the main text, but we have expanded significantly on the discussion of how our self-assembly method allows circumventing the resolution limit of conventional top-down nanofabrication (see SI Section 1) .

2) Secondly, questions arise about the quality and precision of vertical etching. In particular, it is unclear how vertical the etching is for a 200 nm thick Si and how this impacts the final device.

We agree that the verticality of the etching is critical for our self-assembly approach since it determines both whether it happens or not and the final geometry, e.g., the bowtie width. To avoid making this document unnecessarily long, we kindly ask the Reviewer to consult the answer given to Reviewer 2 on the same topic, which can be found below the bolded point that starts as *However, the fact that a gap of about 1 nm [...] an extremely vertical etching profile is required [...]*. The explanations therein have been included in the amended version of the manuscript.

There are also some minor points that should be addressed, such as:

- specifying the color map scale in Fig. 2b

We have specified the colormaps in the revised submission.

- adding a reference to Hu et al., Sci. Adv. 2018;4:eaat2355

We understand that the work of Prof. Sharon Weiss on photonic-crystal nanobeam cavities with dielectric bowties is extensive and, as such, we have included in the main text a reference to Ref. [38]. This work was the first to propose the use of bowtie features exhibiting tightly confined fields as an integral part of the unit cell of a photonic-crystal nanobeam, an approach that we also take here (although with air bowties). We have chosen to cite that reference instead of [Hu et al., Sci. Adv. 2018;4:eaat2355] since we have extensively studied the content of that paper and are determined that it contains a number of fundamental errors. Most importantly, the mode volume could not even in principle be measured in the reported experiment, because the small mode volume in their structure stems from a lightning-rod surface effect inside the structure, i.e., the simulated mode volume is based on the presence of a sharp tip which cannot exist in reality, and does not drop below the diffraction limit above the structure where they attempt to measure it. Also, the calculated mode volume is a numerical artifact and the mode volume in that work is – when doing a properly converged numerical study – close to the diffraction limit. We refer to Ref. [12] for further details.

- improving references to self-assembly and its combination with top-down methods, as in the current version they are relatively outdated.

Following the recommendations of Reviewers 2 and 3, we have improved the references to self-assembly, its combination with top-down methods, and their simultaneous use in MEMS devices. We refer to our response on pages 30-31 of this document for a detailed discussion of these references.

Finally, based on the above concerns, it is unclear how the proposed method can lead to atomic-scale photonic cavities, and the current title seems like an oversell. Instead, it seems like the method has the potential to produce nanometer-to-subnanometer photonic cavities, with the accuracy of the gap still determined by conventional nanolithography steps instead of surface roughness. Therefore, I suggest the authors revise their title and replace "atomic-scale" with "subnanometer" to better reflect their findings. Also, the authors should explicitly clarify the feasibility of a massive scale-up of these bowties in the sub-nanometer regime, or else explain how this resolution is reached and provide experimental evidence for such precision and accuracy on a large scale.

We hope that our explanations above clarify that our title is not an oversell because our method does indeed allow going deep below the resolution limit of conventional lithography and etching (by more than an order of magnitude compared to state-of-the-art CMOS production). We demonstrate devices with atomic-scale features in our manuscript, and our methods are directly applicable to foundry production in which fabrication of features at the scale of a single silicon atom could be realized with a standard deviation well below the size of a single silicon atom. Although our work is research and not production, our new method could in fact – in contrast to many research papers published in the most prestigious journals – be scaled up already today. We note that the title of the amended manuscript has changed to "Self-assembled photonic cavities with atomic-scale confinement" following a comment by Reviewer 2.

-
- [1] Arregui, G. et al. Cavity optomechanics with Anderson-localized optical modes. *Phys. Rev. Lett.* **130**, 043802 (2023).
 - [2] Minkov, M., Dharanipathy, U. P., Houdré, R. & Savona, V. Statistics of the disorder-induced losses of high-Q photonic crystal cavities. *Opt. Express* **21**, 28233–28245 (2013).
 - [3] Eichenfield, M., Chan, J., Camacho, R. M., Vahala, K. J. & Painter, O. Optomechanical crystals. *Nature* **462**, 78–82 (2009).
 - [4] Rosiek, C. A. et al. Observation of strong backscattering in valley-hall photonic topological interface modes. *Nat. Photon.* 1–7 (2023).
 - [5] Loth, F., Kiel, T., Busch, K. & Kristensen, P. T. Surface roughness in finite-element meshes: application to plasmonic nanostructures. *JOSA B* **40**, B1–B7 (2023).
 - [6] Fukuda, A., Asano, T., Kawakatsu, T., Takahashi, Y. & Noda, S. Suppressing the sample-to-sample

- variation of photonic crystal nanocavity Q-factors by air-hole patterns with broken mirror symmetry. *Opt. Express* **31**, 15495–15513 (2023).
- [7] Ramunno, L. & Hughes, S. Disorder-induced resonance shifts in high-index-contrast photonic crystal nanocavities. *Phys. Rev. B* **79**, 161303 (2009).
- [8] Patil, C. M. et al. Observation of slow light in glide-symmetric photonic-crystal waveguides. *Opt. Express* **30**, 12565–12575 (2022).
- [9] Hansen, S. E. et al. Efficient low-reflection fully etched vertical free-space grating couplers for suspended silicon photonics. *Opt. Express* **31**, 17424–17436 (2023).
- [10] Kristensen, P. T., Van Vlack, C. & Hughes, S. Generalized effective mode volume for leaky optical cavities. *Opt. Lett.* **37**, 1649–1651 (2012).
- [11] Albrechtsen, M. et al. Nanometer-scale photon confinement in topology-optimized dielectric cavities. *Nat. Commun.* **13**, 6281 (2022).
- [12] Albrechtsen, M., Vosoughi Lahijani, B. & Stobbe, S. Two regimes of confinement in photonic nanocavities: bulk confinement versus lightning rods. *Opt. Express* **30**, 15458–15469 (2022).
- [13] Hopcroft, M. A., Nix, W. D. & Kenny, T. W. What is the young’s modulus of silicon? *J. Microelectromech. Syst.* **19**, 229–238 (2010).
- [14] Weis, T. A. S., Vosoughi Lahijani, B., Tsoukalas, K., Albrechtsen, M. & Stobbe, S. Design, fabrication, and characterization of electrostatic comb-drive actuators for nanoelectromechanical silicon photonics. Preprint at <https://arXiv:2307.01122> (2023).
- [15] Syms, R. R., Yeatman, E. M., Bright, V. M. & Whitesides, G. M. Surface tension-powered self-assembly of microstructures - the state-of-the-art. *J. Microelectromechanical Syst.* **12**, 387–417 (2003).
- [16] Mastrangeli, M. et al. Self-assembly from milli- to nanoscales: methods and applications. *J. Micromech. Microeng.* **19**, 083001 (2009).
- [17] George, D. & Madou, M. J. In Dixit, U. S. & Dwivedy, S. K. (eds.) *Mechanical Sciences: The Way Forward*, 197–239 (Springer Singapore, 2021).
- [18] Knuesel, R. J. & Jacobs, H. O. Self-assembly of microscopic chiplets at a liquid-liquid-solid interface forming a flexible segmented monocrystalline solar cell. *Proc. Natl. Acad. Sci. U.S.A.* **107**, 993–998 (2010).
- [19] Chang, W. et al. Concurrent self-assembly of RGB microLEDs for next-generation displays. *Nature* **617**, 287–291 (2023).
- [20] Kim, I., Mun, J., Hwang, W., Yang, Y. & Rho, J. Capillary-force-induced collapse lithography for controlled plasmonic nanogap structures. *Microsyst. Nanoeng.* **6**, 1–9 (2020).
- [21] Bajwa, R. & Yapici, M. K. Intrinsic stress-induced bending as a platform technology for controlled self-assembly of high-Q on-chip RF inductors. *J. Micromech. Microeng.* **29**, 064002 (2019).

- [22] Xu, R. & Lin, Y.-S. Flexible and controllable metadvice using self-assembly MEMS actuator. *Nano Lett.* **21**, 3205–3210 (2021).
- [23] Lee, S. W. & Bashir, R. Dielectrophoresis and electrohydrodynamics-mediated fluidic assembly of silicon resistors. *Appl. Phys. Lett.* **83**, 3833–3835 (2003).
- [24] Bishop, K. J., Wilmer, C. E., Soh, S. & Grzybowski, B. A. Nanoscale forces and their uses in self-assembly. *Small* **5**, 1600–1630 (2009).
- [25] Munkhbat, B., Canales, A., Küçüköz, B., Baranov, D. G. & Shegai, T. O. Tunable self-assembled Casimir microcavities and polaritons. *Nature* **597**, 214–219 (2021).
- [26] Takahashi, K., Bulgan, E., Kanamori, Y. & Hane, K. Submicrometer comb-drive actuators fabricated on thin single crystalline silicon layer. *IEEE Trans. Ind. Electron.* **56**, 991–995 (2009).
- [27] Hickey, R., Sameoto, D., Hubbard, T. & Kujath, M. Time and frequency response of two-arm micromachined thermal actuators. *J. Micromech. Microeng.* **13**, 40 (2002).
- [28] Tsoukalas, K., Vosoughi Lahijani, B. & Stobbe, S. Impact of transduction scaling laws on nanoelectromechanical systems. *Phys. Rev. Lett.* **124**, 223902 (2020).
- [29] Klimchitskaya, G., Mohideen, U. & Mostepanenko, V. The Casimir force between real materials: Experiment and theory. *Rev. Mod. Phys.* **81**, 1827 (2009).
- [30] Florez, O. et al. Engineering nanoscale hypersonic phonon transport. *Nat. Nanotechnology* **17**, 947–951 (2022).
- [31] Jansen, H. et al. BSM 7: RIE lag in high aspect ratio trench etching of silicon. *Microelectron. Eng.* **35**, 45–50 (1997).
- [32] Tang, Y., Sandoughsaz, A., Owen, K. J. & Najafi, K. Ultra deep reactive ion etching of high aspect-ratio and thick silicon using a ramped-parameter process. *J Microelectromech Syst.* **27**, 686–697 (2018).
- [33] Ronn, J. et al. Atomic layer engineering of Er-ion distribution in highly doped Er:Al₂O₃ for photoluminescence enhancement. *ACS Photonics* **3**, 2040–2048 (2016).
- [34] Choi, H., Heuck, M. & Englund, D. Self-Similar Nanocavity Design with Ultrasmall Mode Volume for Single-Photon Nonlinearities. *Phys. Rev. Lett.* **118**, 223605 (2017).
- [35] Aoyama, H. et al. The International Roadmap For Devices And Systems (IEEE, 2022); https://irds.ieee.org/images/files/pdf/2022/2022IRDS_Litho.pdf.
- [36] Panuski, C. L. et al. A full degree-of-freedom spatiotemporal light modulator. *Nat. Photon.* **16**, 834–842 (2022).
- [37] Minkov, M., Savona, V. & Gerace, D. Photonic crystal slab cavity simultaneously optimized for ultra-high Q/V and vertical radiation coupling. *Appl. Phys. Lett.* **111**, 131104 (2017).
- [38] Hu, S. & Weiss, S. M. Design of Photonic Crystal Cavities for Extreme Light Concentration. *ACS Photonics* **3**, 1647–1653 (2016).

Reviewer Reports on the First Revision:

Referees' comments:

Referee #1 (Remarks to the Author):

I have considered all the point-by-point answers made from the authors, and the related changes to manuscript. In my opinion, the presentation is certainly improved and the manuscript is now ready to be published in Nature, I have no further comments or issues to raise.

Referee #2 (Remarks to the Author):

The authors have carefully responded to each of my questions and have also revised the main text, and most of the questions I presented previously have been resolved. As a result, I believe that the revised manuscript is worthy of publication in Nature. However, there are two points to be improved.

It is good that the verticality of etching is described on page 4 of the main text and the supporting data is presented in S4.1. However, I think that the influence of this verticality on the gap should be mentioned in the main text. In other words, there should be an explanation that "Such minor sidewall inclinations have a very minor effect on the resulting devices because the compliant halves simply bend by the same angle out of plane, such as to keep the interface parallel". Also, related to Reviewer 3's first comment, I think the statistical average and standard deviation of the gap for devices fabricated with each design (i.e. d) should be mentioned on the main text. The histograms of gaps are shown Fig. S21c but not directly referred in the main text. The average values and standard deviations should be described in the main text because they are important values to guarantee the reproducibility of the gaps. (The average values of the gaps were originally mentioned in the main text, but seems to have been removed in the revised version of the manuscript.)

Referee #3 (Remarks to the Author):

in the revised version of the manuscript, the authors have satisfactorily addressed my comments. I therefore recommend publication.

**Author Rebuttals to First Revision:
Response to the Reviewers' comments on our manuscript:
Self-assembled photonic cavities with atomic-scale confinement.**

Comments by Reviewer#1:

I have considered all the point-by-point answers made from the authors, and the related changes to manuscript. In my opinion, the presentation is certainly improved and the manuscript is now ready to be published in Nature, I have no further comments or issues to raise.

1 September 2023

Our response:

We thank Reviewer#1 for our manuscript's careful and constructive feedback and criticism, which improved the quality of analysis and readability.

Comments by Reviewer#2:

The authors have carefully responded to each of my questions and have also revised the main text, and most of the questions I presented previously have been resolved. As a result, I believe that the revised manuscript is worthy of publication in Nature. However, there are two points to be improved.

It is good that the verticality of etching is described on page 4 of the main text and the supporting data is presented in S4.1. However, I think that the influence of this verticality on the gap should be mentioned in the main text. In other words, there should be an explanation that "Such minor sidewall inclinations have a very minor effect on the resulting devices because the compliant halves simply bend by the same angle out of plane, such as to keep the interface parallel".

Also, related to Reviewer 3's first comment, I think the statistical average and standard deviation of the gap for devices fabricated with each design (i.e. d) should be mentioned on the main text. The histograms of gaps are shown Fig. S21c but not directly referred in the main text. The average values and standard deviations should be described in the main text because they are important values to guarantee the reproducibility of the gaps. (The average values of the gaps were originally mentioned in the main text, but seems to have been removed in the revised version of the manuscript.)

Our response:

We thank Reviewer#2 for the constructive feedback and recommendations to improve the readability. In the revised submission, we have added a sentence following the Reviewer's suggestion to explain the influence of sidewall verticality in the main text (lines 243 to 248). Regarding the mean value and standard deviation of the gaps, we had indeed included them in the main text of the first submission but removed them in our revised submission in order to save space. In the present resubmission, we have put these values back in the main text (lines 273 to 280) along with the standard deviation obtained from Fig. S21c. We would like to note that in the final preparation of our files, we realized that, unfortunately, we had not updated the extracted gaps to reflect the extended analysis (Fig. S21c) of the first resubmission, which means that the average gaps are 0.3 nm smaller than previously stated. We apologize for this mistake, and we note that this has no impact on our findings or claims because although smaller in this case is better, the difference is smaller than the average standard deviation.

Comments by Reviewer#3

In the revised version of the manuscript, the authors have satisfactorily addressed my comments. I therefore recommend publication.

Our response:

We thank Reviewer#3 for the careful and constructive review, which improved the composition and clarity of our manuscript.